# Synergistic correlated states and nontrivial topology in coupled graphene-insulator heterostructures

Xin Lu [1], Shihao Zhang [1], Yaning Wang[2], Xiang Gao[3,4], Kaining Yang[3,4], Zhongqing Guo[1], Yuchen Gao[5,6], Yu Ye [5,6], Zheng Han [3,4] & Jianpeng Liu [1,7] ✉

Graphene has aroused great attention due to the intriguing properties associated with its low-energy Dirac Hamiltonian. When graphene is coupled with a correlated insulating substrate, electronic states that cannot be revealed in either individual layer may emerge in a synergistic manner. Here, we theoretically study the correlated and topological states in Coulomb-coupled and gate-tunable graphene-insulator heterostructures. By electrostatically aligning the electronic bands, charge carriers transferred between graphene and the insulator can yield a long-wavelength electronic crystal at the interface, exerting a superlattice Coulomb potential on graphene and generating topologically nontrivial subbands. This coupling can further boost electron-electron interaction effects in graphene, leading to a spontaneous bandgap formation at the Dirac point and interaction-enhanced Fermi velocity. Reciprocally, the electronic crystal at the interface is substantially stabilized with the help of cooperative interlayer Coulomb coupling. We propose a number of substrate candidates for graphene to experimentally demonstrate these effects.

Graphene hosts two-dimensional (2D) massless Dirac electrons with linear dispersions and nontrivial Berry phases around two inequivalent $K$ and $K'$ valleys in the Brillouin zone (BZ)[1,2]. Such linear dispersions and topological properties of Dirac cones bestow various intriguing single-particle physical properties to graphene including the relativistic Landau levels, the Klein tunneling effects, and the nontrivial edge states, etc.[2]. Besides, low-energy Dirac fermions in graphene also exhibit distinct $e$-$e$ interaction effects[3], such as the interaction-enhanced Fermi velocity[4,5], the gap opening at the charge neutrality point[6–8], and even chiral superconductivity when the Fermi level locates at the van Hove singularity[9].

Insulating transition metal oxides (TMOs) and transition metal chalcogenides (TMCs) have also stimulated significant research interests over the past few decades due to the diverse correlated phenomena discovered in these systems such as Mott insulator[10], excitonic insulator[11,12], and various complex symmetry-breaking states[13,14]. Under charge dopings, these insulating TMOs and/or TMCs may show more intriguing correlated states including unconventional superconductivity[15–17] and long-wavelength charge density wave[18].

An open question is what would happen if two types of distinct interacting many-electron systems, i.e., the interacting Dirac fermions in graphene and the correlated electrons in (slightly) charge doped

[1]School of Physical Science and Technology, ShanghaiTech University, Shanghai 201210, China. [2]Shenyang National Laboratory for Materials Science, Institute of Metal Research, Chinese Academy of Sciences, Shenyang, China. [3]State Key Laboratory of Quantum Optics and Quantum Optics Devices, Institute of Opto-Electronics, Shanxi University, 030006 Taiyuan, China. [4]Collaborative Innovation Center of Extreme Optics, Shanxi University, 030006 Taiyuan, China. [5]Collaborative Innovation Center of Quantum Matter, Beijing 100871, China. [6]State Key Lab for Mesoscopic Physics and Frontiers Science Center for Nano-Optoelectronics, School of Physics, Peking University, Beijing 100871, China. [7]ShanghaiTech Laboratory for Topological Physics, ShanghaiTech University, Shanghai 201210, China. ✉e-mail: liujp@shanghaitech.edu.cn

TMO and/or TMC insulators, are integrated into a single platform. Especially, how the mutual couplings would affect the interacting electronic states in both systems. Inspired by recent pioneering experiments in CrOCl-graphene[19], 1T-TaS$_2$-graphene[20], and CrI$_3$-graphene[21] heterostructures, here we propose that such a scenario (of interacting Dirac fermions coupled with the correlated electrons in charge doped TMO/TMC insulators) can be realized in graphene-insulator heterostructures with gate-tunable band alignment. In this work, we show that, by virtue of the interlayer Coulomb coupling between the interacting electrons in the two layers, intriguing correlated physics that cannot be seen in either individual layer would emerge in a cooperative and synergistic manner in such band-aligned graphene-insulator heterostructures.

When Dirac points of graphene are energetically close to the band edge of the insulating substrate, charge carriers can be transferred between graphene and the substrate under the control of gate voltages due to quantum tunneling effects. This may yield a long-wavelength electronic crystal (EC) at the surface of the substrate, given that the carrier density introduced to the substrate is below a threshold value. On the one hand, the long-wavelength EC at the surface of the substrate would impose an interlayer superlattice Coulomb potential to graphene, which would generate subbands with reduced non-interacting Fermi velocity of the Dirac cone, thus trigger gap opening at the Dirac points by *e-e* interactions in graphene. Meanwhile, concomitant with the gap opening, the Fermi velocities around the charge neutrality point (CNP) are dramatically enhanced due to *e-e* interactions effects. The subbands may also possess nontrivial topological properties with non-zero valley Chern numbers that can be controlled by superlattice constant and anisotropy. Especially, we find a number of "magic lines" in the parameter space of superlattice's constant and anisotropy, at which the Fermi velocity along one direction vanishes exactly. The subbands would acquire non-zero Chern numbers when passing through these magic lines. On the other hand, the gapped Dirac state at the CNP of graphene would further

stabilize the long-wavelength EC state in the substrate by pinning the relative charge centers of the two layers in an anti-phase interlocked pattern, in order to optimize the interlayer Coulomb interactions.

## Results

### Coulomb interaction effects in graphene

To describe the graphene-insulator heterostructure, we consider a model Hamiltonian consisted of a graphene part, an insulator substrate part, and the coupling between them (see Eqs. (5) and Supplementary Note 6 of Supplementary Information). As we are interested in the low-energy electronic properties, graphene's band structure is modeled by the low-energy Dirac cones around the $K$ and $K'$ valleys. The long-wavelength EC (charge ordered) state in the substrate is considered as a charge insulator, with the electrons being frozen in the form of a superlattice, as schematically shown in Fig. 1a. Thus, long-wavelength charge order of the substrate is coupled to the graphene layer via interlayer Coulomb interactions to exert a superlattice potential on the Dirac electrons (see Fig. 1b). If one considers that the charge order in the substrate layer results from a Wigner-crystal-like instability, then the value of superlattice constant $L_s = 50$ Å would correspond to a carrier density ~ $7 \times 10^{12}$ cm$^{-2}$ transferred from graphene to the insulating substrate, which is close to the upper limit for a double-gated graphene device. Neglecting the intervalley coupling thanks to the large superlattice constant $L_s$ ($\gtrsim$50 Å), we can construct an effective single-particle Hamiltonian for the continuum Dirac fermions in graphene that are coupled with a superlattice Coulomb potential (see Supplementary Note 1 and Supplementary Note 6 in Supplementary Information)

$$H_0^\mu(\mathbf{r}) = \hbar v_F \mathbf{k} \cdot \boldsymbol{\sigma}^\mu + U_d(\mathbf{r}) \tag{1}$$

where $\boldsymbol{\sigma}^\mu$ are the Pauli matrices ($\mu\sigma_x, \sigma_y$) with the valley index $\mu = \pm 1$, $v_F$ is the non-interacting Fermi velocity of graphene, and $U_d(\mathbf{r})$ is the background superlattice potential with the period $U_d(\mathbf{r}) = U_d(\mathbf{r} + \mathbf{L_s})$.

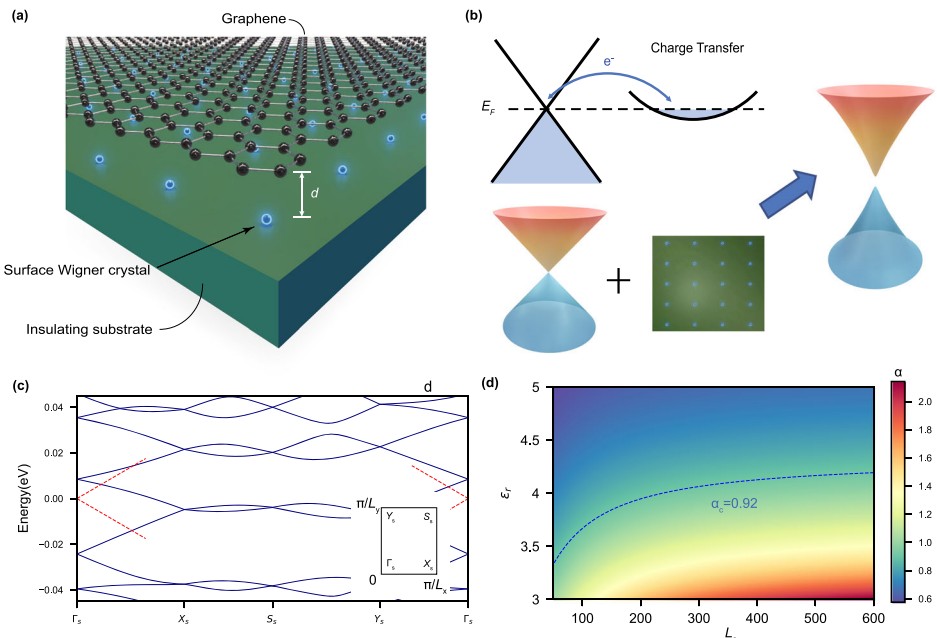

**Fig. 1 | Mechanism of enhanced *e-e* interaction effects by virtue of charge transfer in graphene-insulator heterostructure. a** Cartoon illustration of a monolayer graphene (black balls with sticks) supported by an insulating substrate (green platform) with long-wavelength charge order (blue dots), with an interlayer distance *d*. **b** Schematic of charge transfer in a band-aligned graphene-insulator heterostructure and its effects on the Dirac dispersion. The dashed line marks the Fermi energy $E_F$. **c** Blue lines show the non-interacting band structure of graphene

coupled to a superlattice Coulomb potential with rectangular symmetry. The anisotropy parameter is $r = 1.2$ and the superlattice constant $L_x = L_s = 600$ Å. The red dashed lines represent the non-interacting Dirac cones in free-standing graphene. The inset marks the high-symmetry points in the superlattice Brillouin zone. **d** The calculated effective fine structure constant $\alpha$ as a function of $L_s$ and dielectric constant $\epsilon_r$, where the dashed line marks the critical value $\alpha_c \approx 0.92$.

The superlattice of the EC is set to be rectangular, with anisotropy $r = L_y/L_x$ and $L_{x,y}$ being the superlattice constant in the $x, y$-direction, respectively. We denote $L_s = L_x$. As a result, the superlattice potential $U_d(\mathbf{r})$ would fold Dirac cones into its small Brillouin zone, forming subbands and opening up a gap at the boundary of the supercell BZ, as shown in Fig. 1c for a rectangular superlattice with $r = 1.2$ (same as that of CrOCl atomic lattice) in valley $K$ ($\mu = 1$) with $L_s = 600$ Å. The energy degeneracies from folding are all lifted by $U_d$, whose Fourier component reads

$$U_d(\mathbf{Q}) = \frac{e^2}{\epsilon_0 \epsilon_r \Omega_0} \frac{e^{-|\mathbf{Q}|d}}{|\mathbf{Q}|}, \qquad (2)$$

where $\mathbf{Q} \neq \mathbf{0}$ is the reciprocal lattice vector associated with $\mathbf{L}_s$, $\Omega_0 = L_x L_y$ is the area of the primitive cell of the superlattice. The Coulomb potential $U_d$, screened by a dielectric constant $\epsilon_r$, decays exponentially in the reciprocal space $\sim \exp(-Qd)$, where $d$ is the distance between the substrate surface and graphene monolayer. Furthermore, the Fermi velocities near the Dirac points of the subbands are suppressed by $U_d$[22] as clearly shown in Fig. 1c. Such a continuum-model description is adopted throughout the paper given that $L_s \gg a$ ($a = 2.46$ Å is graphene's lattice constant) is always fulfilled for low carrier density $n \lesssim 10^{13}$ cm$^{-2}$, with $L_s \sim 1/\sqrt{n}$ for the EC state.

While it is highly desirable to open a gap at the Dirac points in graphene for the purpose of field-effect device fabrication, the superlattice potential of Eq. (2) alone cannot gap out Dirac points in graphene as the system still preserves $C_{2z}\mathcal{T}$ symmetry. However, the Dirac points can be unstable against $e$-$e$ Coulomb interactions (with the spontaneous breaking of $C_{2z}\mathcal{T}$ symmetry) once the Fermi velocity of the non-interacting band structure is suppressed below a threshold, which can be assisted by the superlattice potential from the long-wavelength charge order. One of the similar illustrations is twisted bilayer graphene (TBG)[23], where the Fermi velocity is strongly suppressed around the "magic angle", leading to moiré flat bands exhibiting diverse correlated and topological phases[24–29]. Here we further calculate the Fermi velocity of the superlattice subbands around the Dirac point, denoted as $v_F(L_s, \epsilon_r)$, which depends on both the superlattice constant $L_s$ and the background dielectric constant $\epsilon_r$. Accordingly, the effective fine structure constant $\alpha(L_s, \epsilon_r) = e^2/(4\pi\epsilon_0\epsilon_r\hbar v_F(L_s, \epsilon_r))$ can also be tuned by $L_s$ and $\epsilon_r$, as shown in Fig. 1d. We see that there is a substantial region in the $(L_s, \epsilon_r)$ parameter space with $\alpha(L_s, \epsilon_r) > \alpha_c \approx 0.92$[30], which indicates that the Dirac-semimetal phase of graphene may no longer be stable against $e$-$e$ interactions within this regime according to previous theoretical study[30].

Such a picture is not unique to rectangular superlattice, but applies to various superlattice geometries. Treating the superlattice potential $U_d(\mathbf{Q})$ using second-order perturbation theory, the renormalized non-interacting effective Hamiltonian for arbitrary superlattice geometry can be expressed as

$$H_{\text{eff}}^0(\mathbf{k}) = \hbar v_F \left( 1 - \sum_{|\mathbf{Q}| \neq 0} \frac{|U_d(\mathbf{Q})|^2}{(\hbar v_F)^2 |\mathbf{Q}|^2} \right) \left( \mathbf{k} - \sum_{|\mathbf{Q}| \neq 0} \frac{|U_d(\mathbf{Q})|^2}{(\hbar v_F)^2 |\mathbf{Q}|^2} \left( \mathbf{k} - \frac{2\mathbf{k} \cdot \mathbf{Q}}{|\mathbf{Q}|^2} \mathbf{Q} \right) \right) \cdot \boldsymbol{\sigma}. \qquad (3)$$

We see that the effective non-interacting Hamiltonian as well as the Fermi velocity have similar dependence on $L_s$ and $\epsilon_r$ (through $U_d(\mathbf{Q})$) for all lattice geometries. We have also calculated the effective fine-structure constants $\alpha(L_s, \epsilon_r) = e^2/(4\pi\epsilon_0\epsilon_r\hbar v_F(L_s, \epsilon_r))$ for both triangular and square lattices (see Supplementary Figure 2), and the results are very similar to that of rectangular lattice with $r = 1.2$ shown in Fig. 1d.

This motivates us to include $e$-$e$ interactions in the graphene layer in our model. Despite several theoretical predictions of gapped Dirac states in graphene[3,6–8,31], to the best of our knowledge no gap at the CNP has been experimentally observed in suspended graphene yet[32,33]. This can be attributed to interaction-enhanced Fermi velocity around

the CNP, screening of $e$-$e$ interactions due to ripple-induced charge puddles, disorder effects, etc.[3,34–37]. Nevertheless, analogous to TBG, the subbands in our system with reduced non-interacting Fermi velocity would quench the kinetic energy and further promote the $e$-$e$ interaction effects in graphene.

Our unrestricted Hartree-Fock calculations (see Supplementary Note 4 in Supplementary Information) confirm precisely the argument above. As interaction effects are most prominent around the CNP, we project the Coulomb interactions onto only a low-energy subspace including three valence and three conduction subbands ($n_{\text{cut}} = 3$) that are closest to CNP for each valley and spin. To incorporate the influences of Coulomb interactions from the high-energy remote bands, the renormalized Fermi velocity within the low-energy subspace can be derived from the renormalization group (RG) approach[2–4,38]

$$v_F^* = v_F \left( 1 + \frac{\alpha_0}{4\epsilon_r} \log \frac{E_c}{E_c^*} \right), \qquad (4)$$

where $\alpha_0 = e^2/(4\pi\epsilon_0\hbar v_F)$ is the ratio between the Coulomb interaction energy and kinetic energy, i.e., the effective fine-structure constant of free-standing graphene, $E_c^*$ delimits the low-energy window within which the unrestricted Hartree-Fock calculations are to be performed, and $E_c$ is an overall energy cut-off above which the Dirac-fermion description to graphene is no longer valid. Unlike TBG[39], other parameters of the effective Hamiltonian (Eq. (1)) such as $U_d$, are unchanged under the RG flow (see Supplementary Note 3 in Supplementary Information).

We first study the interaction effects of graphene coupled to a rectangular superlattice potential with $r = 1.2$ and $50$ Å $\leq L_s \leq 400$ Å, corresponding to carrier density of the EC state at the surface of the substrate $0.1 \times 10^{12}$ cm$^{-2} \leq n \leq 6.58 \times 10^{12}$ cm$^{-2}$ (with $n = 2/(rL_s^2)$), with $\epsilon_r = 3, 4$, and $d = 7$ Å (obtained from first principles density functional theory calculations for one particular commensurate CrOCl-graphene supercell (see Supplementary Note 7 in Supplementary Information)). Here, we consider two different filling factors: exactly at the CNP ($\nu = 0$) and a slight hole doping ($\nu \approx -0.003$). When $\nu = 0$, a gap can be opened up due to interaction effects (see Fig. 2a, b), leading to two nearly degenerate insulating states, one is $\sigma_z$-sublattice polarized and the other is characterized by the order parameter $\tau_z\sigma_z$, where $\tau_z$ and $\sigma_z$ denote the third Pauli matrix in valley and sublattice space, respectively. Then, intervalley Coulomb interactions would split such degeneracy, and the sublattice polarized insulator with zero Chern number becomes the unique ground state (see Supplementary Note 5 in Supplementary Information). Notably, the gap decreases almost linearly with $n$ as shown in Fig. 2d, and eventually vanishes as $n \to 0$. This is because the superlattice Coulomb potential exerted on graphene is proportional to the carrier density of the long-wavelength order from the substrate. Consequently, the Fermi velocity of the bare Dirac dispersion of graphene would be less suppressed at smaller carrier density $n$, which disfavors gap opening. Eventually in the limit of $n \to 0$, with a charge ordered state of infinite lattice constant, graphene would recover its non-interacting behavior as a gapless Dirac semimetal.

To verify our theory, we have also experimentally measured the gaps at CNP in graphene-CrOCl heterostructure at different nominal carrier densities using the same high-quality device reported in ref. 19. The details for the measurement set up and the device configuration are presented in Supplementary Note 8 of Supplementary Information. The measured gaps also decrease linearly with $n_{\text{tot}}$, from 7.7 meV with $n_{\text{tot}} = 3.4 \times 10^{12}$ cm$^{-2}$, to 5.8 meV with $n_{\text{tot}} = 0.5 \times 10^{12}$ cm$^{-2}$ (see Supplementary Figure 20), consistent with the trend from theoretical calculations, as shown in Fig. 2e. Nevertheless, when $n_{\text{tot}} \to 0$, such a linear dependence of the gap on $n_{\text{tot}}$ may no longer be valid. This is because in Eq. (2), the interlayer Coulomb potential only applies to the situation of a single valley to accommodate charge carriers in the substrate. In

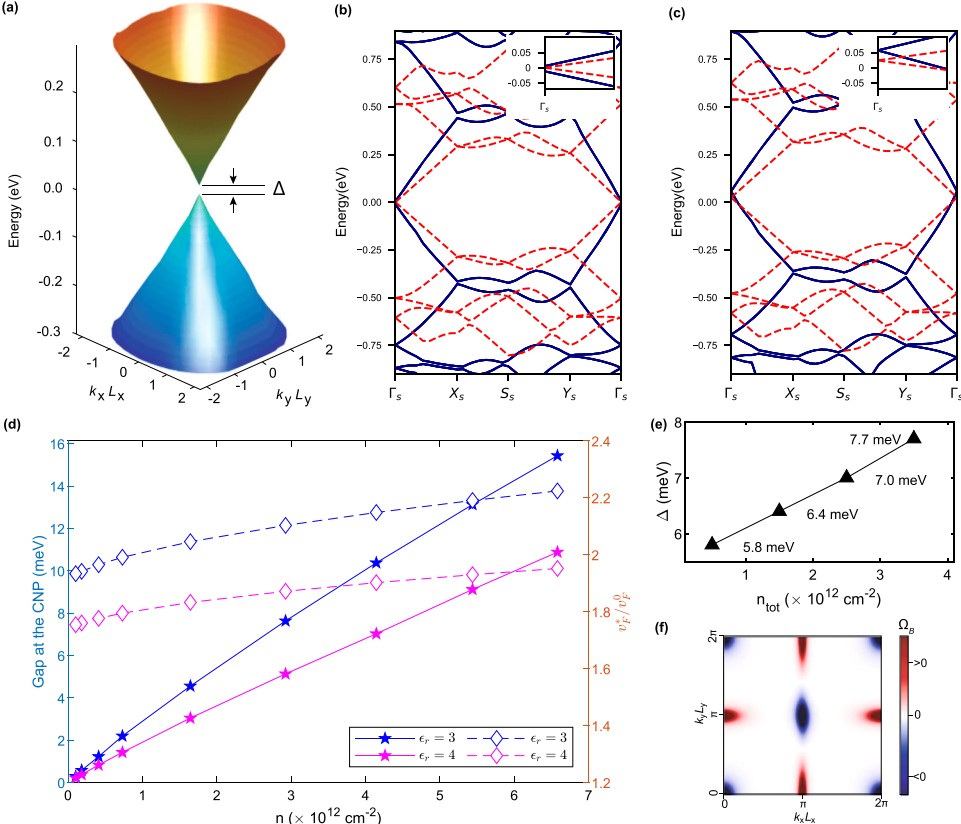

**Fig. 2 | Gap opening and enhanced Fermi velocity in graphene coupled with a superlattice Coulomb potential. a** Calculated Hartree-Fock single-particle excitation spectrum of graphene coupled to a superlattice Coulomb potential as a function of dimensionless reduced momenta $(k_x L_x, k_y L_y)$, with the filling set at charge neutrality point. A gap $\Delta$ is opened at the Dirac point. **b** and **c** show by blue solid lines the Hartree-Fock band structures of $L_s = 50$ Å and $\epsilon_r = 3.0$, with the filling factor $v = 0$ in (**b**) and $v = -0.003$ in (**c**). The red dashed lines represent the non-interacting Dirac cones. The insets zoom in energy close to the Dirac points. Zero energies in (**b**) and (**c**) are defined as the chemical potentials for $v = 0$ and $v = -0.003$, respectively. **d** The calculated gaps at charge neutrality point (filled stars) and the interaction-enhanced Fermi velocities at slight hole dopings $v = -0.003$ (hollow diamonds) as a function of the substrate's carrier density $n$. **e** The thermal activation gap $\Delta$ measured on the devices in ref. 19 for different nominal dopings $n_{tot}$. **f** Distribution of Berry curvature $\Omega_B$ of the highest valence subband of $K$ valley for $r = 1.2$ and $L_s = 50$ Å, which gives zero valley Chern number.

reality, there may be additional valley degeneracy in the substrate, which is crucial for the evolution of gap as $n_{tot} \to 0$. Although the valley degeneracy of the substrate does not change our results qualitatively, the theoretically calculated gap vs. $n_{tot}$ fits to the experimental data of CrOCl-graphene heterostructure more precisely at low density once including the two-fold valley degeneracy of CrOCl (see Table 1). The details are given in Supplementary Note 5 in Supplementary Information.

We note that the electronic crystal at the surface of the substrate is expected to persist even if the carrier density exceeds the threshold value due to the extra energy gain from interlayer Coulomb coupling in such coupled system, which will be discussed in detail in the subsection "Cooperative coupling between graphene and substrate" below. Strain is also inevitable in such graphene-insulator heterostructures, which would give rise to pseudo-magnetic fields coupled to the Dirac electrons[5,40,41], thus further enhance the *e-e* interaction effects in graphene.

The single-particle excitation spectrum is also significantly altered by Coulomb interactions within the low-energy window, as shown in Fig. 2b and c with fillings $v = 0$ and $v = -0.003$, respectively. We note that although the superlattice potential $U_d$ suppresses Fermi velocity in graphene (see Fig. 1c), *e-e* interactions can compensate such effects. The Fermi velocity is not only enhanced by the Coulomb potentials from the remote energy bands (Eq. (4)), but also further boosted by *e-e* interactions within the low energy

**Table 1 | Candidate substrate materials for the graphene-insulator heterostructure systems**

| Materials | $\epsilon_r$ | $E_{CBM}$ | $E_{VBM}$ | $m^*/m_O$ | $g_v$ | $r_s$ |
|---|---|---|---|---|---|---|
| AgScP$_2$S$_6$ (bi) | 3.67 | 0.07 eV | −1.89 eV | 3.94 | 6 | 683.4 |
| AgScP$_2$Se$_6$ (bi) | 4.06 | 0.15 eV | −1.37 eV | 2.63 | 6 | 412.8 |
| IrBr$_3$ (bi) | 6.53 | 0.23 eV | −1.43 eV | 8.08 | 2 | 262.7 |
| IrI$_3$ (bi) | 7.59 | 0.33 eV | −0.95 eV | 1.76 | 2 | 49.1 |
| YI$_3$ (tri) | 3.45 | 0.53 eV | −2.1 eV | 2.12 | 1 | 65.3 |
| YBr$_3$ (tri) | 6.78 | 0.68 eV | −3.15 eV | 2.76 | 1 | 43.3 |
| ReSe$_2$ (bi) | 6.38 | 0.32 eV | −0.83 eV | 1.82 | 2 | 60.7 |
| ScOCl (bi) | 5.27 | 0.21 eV | −4.04 eV | 3.29 | 1 | 66.2 |
| PbO (bi) | 8.47 | 2.02 eV | −0.03 eV | 11.89 | 4 | 595.8 |
| CrI$_3$ (bi) | 3.00 | −0.32 eV | −1.58 eV | 2.02 | 2 | 142.8 |
| CrOCl (bi) | 3–4 | −0.13 eV | −3.26 eV | 1.31 | 2 | 55.7–74.2 |
| WS$_2$ (tri,quad) | 3.63 | 0–0.08 eV | −1.01 – −0.97 eV | 1.16 | 6 | 201–203 |
| WSe$_2$ (tri,quad) | 4.07 | 0.27–0.47 eV | −0.65 – −0.52 eV | 0.53 | 6 | 87.4 |
| MoSe$_2$ (bi, tri, quad) | 7.29 | −0.01–0.31 eV | −0.97 – −0.86 eV | 0.73–0.77 | 6 | 66–70 |
| MoTe$_2$ (bi, tri, quad) | 6.75 | 0.31–0.42 eV | −0.54 – −0.47 eV | 0.7–0.75 | 6 | 68–73 |

The dielectric constants $\epsilon_r$,[64–66] conduction band minimum position ($E_{CBM}$), valence band maximum position ($E_{VBM}$), the corresponding effective mass $m^*$ at the band edge that is closer to the Dirac point (set to zero) in energy, and the dimensionless Wigner-Seitz radius $r_s = g_v m^*/\sqrt{\pi n}\epsilon_r m_O a_B^0$ ($a_B^0$ is the Bohr radius and $m_O$ is the bare electron mass, $g_v$ is the valley degeneracy) estimated under a small doping concentration $n = 10^{12}$ cm$^{-2}$, are presented. Here bi and tri stand for bilayer and trilayer systems, respectively.

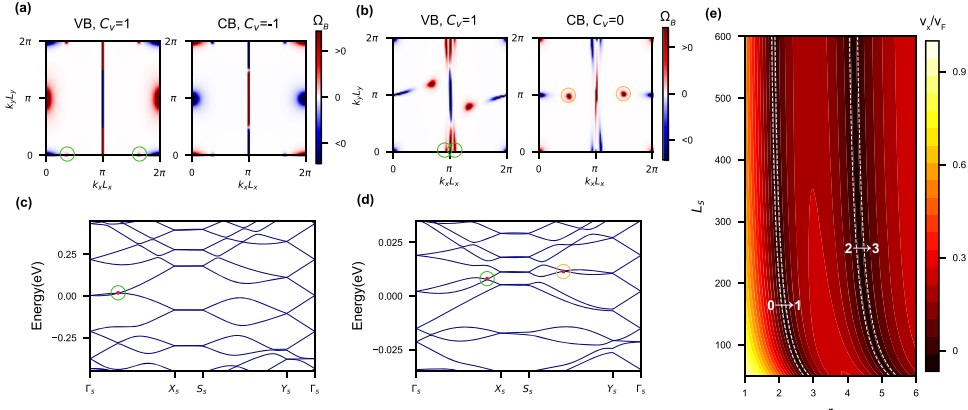

**Fig. 3 | Nontrivial topological properties controlled by the superlattice's lattice constant and anisotropy. a, b** The distribution of Berry curvature $\Omega_B$ in the $r = 3$ superlattice's Brillouin zone (BZ) of the lowest valence band (VB) and conduction band (CB) in valley $K$ for $L_s = 50$ and 600 Å, respectively. Their corresponding valley Chern number $C_v$ is also given at the top of each panel. **c, d** The non-interacting band structure of the $r = 3$ superlattice with $L_s = 50$ and 600 Å. The green and orange circles in (**a**) and (**b**) indicate spots in the BZ with highly

concentrated Berry curvatures, which cause the topological transitions. These spots are generated by band inversion points circled in (**c**) and (**d**) using the same colors as in (**a**) and (**b**). **e** Color map of Fermi velocity in the $x$-direction $v_x$ of the valence band for $\epsilon_r = 3$. The color coding indicates $v_x/v_F$. Here we vary $L_x$ from 50 to 600 Å and anisotropy parameter $r$ from 1 to 6. The white dashed line, i.e., the "magic lines", mark the position in parameter space where $v_x$ vanishes.

window $E_c^* \sim n_{cut}\hbar v_F 2\pi/L_s$. Eventually, the Fermi velocity can be magnified up to more than twice of the non-interacting value of free-standing graphene ($v_F$) at slight hole doping $\nu = -0.003$, as shown in Fig. 2d. This explains the recent experiment in gate-controlled graphene-CrOCl heterostructure, in which the Fermi velocity around CNP is significantly enhanced compared to non-interacting value at slight carrier doping, such that robust quantum Hall effect can be observed under tiny vertical magnetic fields (~0.1 T) and at high temperatures[19]. We note that the EC state may be stabilized by vertical magnetic fields even when the carrier density in the substrate exceeds the zero-field threshold value[42,43], which in turn boosts the low-field, high-temperature quantum Hall effect in the graphene layer due to the scenario discussed above.

Although it has been theoretically proposed that the magnetic proximity effect together with spin-orbit coupling could in principle give rise to topologically nontrivial states in graphene[44], it seems to be irrelevant to the graphene-insulator heterostructures considered in the present study. For example, in CrOCl-graphene device, no magnetic hysteresis has been observed in graphene, and the measured Landau level degeneracy is still compatible with that of spin-valley degenerate Dirac cones[19]. Most saliently, the gap opening and the robust quantum Hall effect persist up to temperatures far above the Néel temperature of CrOCl (~14 K)[19]. Similarly, the magnetic proximity coupling was also reported to be negligible for CrI₃-graphene heterostructure[21]. Therefore, compared to the power-law decaying interlayer Coulomb coupling, the exponentially decaying magnetic proximity coupling may not play an important role in such charge-transfer graphene-insulator heterostructures.

The essential results discussed above, i.e., the gap opening at CNP and the concomitant drastic enhancement of Fermi velocity, remain valid for different types of the background superlattices. Specifically, we have also performed calculations for the case of triangular superlattices, which lead to qualitatively the same conclusions, as presented in Supplementary Note 5 of Supplementary Information.

## Topological properties

Different from magic-angle TBG[45–49], the low-energy subbands for graphene coupled to a rectangular superlattice potential $U_d(\mathbf{r})$ with small anisotropy ($r \sim 1$) turn out to be topologically trivial with a compensating Berry-curvature distribution, leading to zero Chern number. This remains true even in the gapped Dirac state after including $e$-$e$

interactions, as shown in Fig. 2f. The trivial band topology is somehow anticipated because the superlattice potential is non-chiral in the sense that it is coupled equally to the two sublattice of graphene, which does not have any pseudo-gauge-field structure such as that in TBG[49,50].

Hence, it is unexpected that changing the anisotropy $r$ and the lattice size $L_s$ of the superlattice potential $U_d$ can make the subbands topological. For example, keeping $L_x = 50$ Å but with $r = 3.0$, both the highest valence band and the lowest conduction band acquire non-zero valley Chern numbers $C_v = \pm 1$ (after adding an infinitesimal $C_{2z}$-breaking staggered sublattice potential). As shown in Fig. 3a, besides the four high-symmetry points, it appears another two hot spots with concentrated Berry curvatures (annotated by green circles) along the line connecting $\Gamma_s$ and $X_s$. This additional contribution breaks the balance between positive and negative contribution of Berry curvature to Chern number, leading to non-zero valley Chern number. Such contribution stems from another accidental crossing point between the low-energy valence and conduction bands along the $k_x$-direction through changing merely the anisotropy parameter $r$, as shown in Fig. 3c by red dot within green circle.

While increasing $r$ from unity (with fixed $L_s$), the Fermi velocity in the $x$-direction of the valence band around the Dirac point, $v_x$, is gradually reduced, as shown in Fig. 3e. As the same origin of Klein tunneling effects, the spinor structure of graphene's wavefunction forces the Fermi velocity in the $y$-direction to be intact[22]. Further tuning $r$ at some point would totally flatten $v_x$. In Fig. 3e, we mark by white dashed lines "the magic lines" on which $v_x$ of the valence band closest to Dirac points vanishes exactly. The magic lines always come in pair as an effect of chiral (particle-hole) symmetry breaking induced by the superlattice potential. As particle-hole symmetry is broken in the energy spectrum, when $v_x$ vanishes in the valence band, the counterpart in the conduction band remains finite. The valence subband around the Dirac point has to curve upwards to create an accidental band crossing point, after that $v_x$ of the valence band becomes zero again. Therefore, a band crossing would be germinated at the Dirac point, and then move away along the $k_x$-direction with larger $r$. On the one hand, the band crossing moving away from $\Gamma_s$ is of accidental nature, which is generally avoided unless the lattice parameters are at some fine-tuned values. On the other hand, the Dirac point at $\Gamma_s$ remains stable as protected by $C_{2z}\mathcal{T}$ symmetry. If the Dirac point is gapped, say, by a tiny staggered sublattice potential, the low-energy

subbands become topological with non-zero valley Chern numbers. In particular, with the increase of $r$ at fixed $L_s$, the absolute value of valley Chern number of the valence subband (closest to Dirac points) increases by 1 whenever one pair of the magic lines are passed through. The positions of these magic lines also depend on the background dielectric constant $\epsilon_r$ since larger $\epsilon_r$ corresponds to weaker Fermi-velocity renormalization effect, which would shift the magic lines to larger $r$ values. In Supplementary Information, we provide animated figure (Supplementary Figure 4) and videos (Supplementary Movies 1–6) demonstrating the evolution of the band structures and Berry curvatures with increasing $r$ at fixed $L_s$. Such topologically nontrivial subbands with highly anisotropic Fermi velocities may provide an alternative platform to realize topological quantum matter.

We note that the anisotropic charge ordered superlattices may be realized in two ways. First, one can design a spatially modulated electrostatic potential, which has been realized in monolayer graphene by inserting a patterned dielectric superlattice between the gate and the sample[51]. Then, the anisotropy of the superlattice can be artificially tuned by the dielectric patterning in the substrate. Second, for some given carrier density, the Fermi surface of the conduction (or valence) band of the substrate may be (partially) nested, which may lead to a charge density wave (CDW) state with the nesting wavevector. For example, for CrOCl, the Fermi surfaces under different Fermi energies (above the conduction band minimum) are given in Supplementary Figure 15c. Clearly, under some proper fillings, the Fermi surfaces are nested or partially nested, which may give rise to CDW states with anisotropic superlattices. We note that topologically nontrivial flat bands have also been proposed to exist in Bernal bilayer graphene coupled with a background superlattice potential[52].

Furthermore, we find that changing $L_s$ is also able to control the valley Chern number of the subbands. For example, with $r = 3$ and $L_s = 600$ Å, as shown in Fig. 3b, while the highest valence band remains topological with non-zero valley Chern number 1 for valley $K$ with the two aforementioned crossing points (green circles) merely moving to $X_s$, the lowest conduction band turns out to be topologically trivial. This is due to two additional band crossing points (orange circles) close to the $Y_s$-$S_s$ line between the lowest and the second lowest conduction bands, as annotated by red dots in an orange circle in Fig. 3d.

The nontrivial topology must arise from the intrinsic Berry phases of the Dirac cones. Such topologically nontrivial bands are particularly surprising for our system, since the Dirac fermions are subjected to a trivial superlattice potential, which couples identically with two sublattices of graphene. Nevertheless, the nontrivial subband topology is highly tunable by changing the superlattice's size and anisotropy (see Supplementary Note 2 in Supplementary Information).

**Cooperative coupling between graphene and substrate**
In the previous calculations, a charge ordered superlattice in the substrate is presumed, which exerts a classical superlattice Coulomb potential to graphene. However, this assumption should be re-examined. Moreover, besides the effects from the substrate to graphene, the feedback effects from graphene to the substrate should be discussed as well. Therefore, in this section, we study the coupled bilayer system as a whole, and treat the electrons in graphene layer and the substrate layer on equal footing. In particular, we model the carriers transferred to the substrate as 2D electron gas with long-range $e$-$e$ Coulomb interactions. Electrons in the substrate and in graphene interact with each other via long-range Coulomb potential, whose Fourier component of wavevector $\mathbf{q}$ reads $e^2 \exp(-|\mathbf{q}| d)/(2\epsilon_0 \epsilon_r |\mathbf{q}|)$. Thus, the total Hamiltonian for the Coulomb-coupled graphene-

insulator heterostructure system includes:

$$H_{\text{gr}}^0 = \sum_{\mathbf{k},\mu,\alpha,\alpha',\sigma} (\hbar v_F \mathbf{k} \cdot \boldsymbol{\sigma}^\mu)_{\alpha,\alpha'} \hat{c}_{\sigma\mu\alpha}^\dagger(\mathbf{k}) \hat{c}_{\sigma\mu\alpha'}(\mathbf{k}), \tag{5a}$$

$$H_{\text{sub}}^0 = \sum_{\mathbf{k},\sigma} \left( \frac{\hbar^2 \mathbf{k}^2}{2m^*} + E_{\text{CBM}} \right) \hat{d}_\sigma^\dagger(\mathbf{k}) \hat{d}_\sigma(\mathbf{k}), \tag{5b}$$

$$H_{\text{gr}}^{\text{intra}} = \frac{1}{2S} \sum_{\substack{\sigma,\sigma' \\ \mu,\mu'}} \sum_{\substack{\alpha,\alpha' \\ \mathbf{k},\mathbf{k}',\mathbf{q}}} V_{\text{int}}(\mathbf{q}) \hat{c}_{\sigma\mu\alpha}^\dagger(\mathbf{k}+\mathbf{q}) \hat{c}_{\sigma'\mu'\alpha'}^\dagger(\mathbf{k}'-\mathbf{q}) \hat{c}_{\sigma'\mu'\alpha'}(\mathbf{k}') \hat{c}_{\sigma\mu\alpha}(\mathbf{k}), \tag{5c}$$

$$H_{\text{sub}}^{\text{intra}} = \frac{1}{2S} \sum_{\mathbf{k},\mathbf{k}',\mathbf{q}} \sum_{\sigma,\sigma'} V_{\text{int}}(\mathbf{q}) \hat{d}_\sigma^\dagger(\mathbf{k}+\mathbf{q}) \hat{d}_{\sigma'}^\dagger(\mathbf{k}'-\mathbf{q}) \hat{d}_{\sigma'}(\mathbf{k}') \hat{d}_\sigma(\mathbf{k}), \tag{5d}$$

$$H_{\text{gr-sub}} = \frac{1}{S} \sum_{\mu,\alpha,\sigma,\sigma'} \sum_{\mathbf{k},\mathbf{k}',\mathbf{q}} \frac{e^2 e^{-|\mathbf{q}|d}}{2\epsilon_0 \epsilon_r |\mathbf{q}|} \hat{c}_{\sigma\mu\alpha}^\dagger(\mathbf{k}) \hat{d}_{\sigma'}^\dagger(\mathbf{k}') \hat{d}_{\sigma'}(\mathbf{k}'-\mathbf{q}) \hat{c}_{\sigma\mu\alpha}(\mathbf{k}+\mathbf{q}). \tag{5e}$$

On the graphene side, Eq. (5a) is the familiar Dirac Hamiltonian describing the non-interacting low-energy physics of graphene. The $e$-$e$ Coulomb interactions within graphene are described by Eq. (5c), where the dominant intravalley long-range Coulomb interactions are considered and $V_{\text{int}}(\mathbf{q})$ is in the form of double-gate screened Coulomb potential (see Eq. (9)). Here, $\hat{c}_{\sigma\mu\alpha}(\mathbf{k})$ and $\hat{c}_{\sigma\mu\alpha}^\dagger(\mathbf{k})$ denote annihilation and creation operators for the low-energy Dirac electrons with wavevector $\mathbf{k}$, valley $\mu$, spin $\sigma$, and sublattice $\alpha$. Note that $S$ refers to the total surface area of the coupled system, and the atomic wavevectors $\mathbf{k},\mathbf{k}',\mathbf{q}$ are expanded around the Dirac points. On the substrate side, without loss of generality, we suppose that the chemical potential is close to the conduction band minimum (CBM) with its energy $E_{\text{CBM}}$, and the energy dispersion of the low-energy electrons around CBM can be modeled by a parabolic band as for 2D free electron gas with effective mass $m^*$. Other electrons in the deep valence bands are supposed to be integrated into the static dielectric screening constant thanks to a large gap of the substrate. Therefore, the non-interacting Hamiltonian Eq. (5b) for electrons in the substrate can be written in the plane wave basis with creation and annihilation operators $\{\hat{d}_\sigma^\dagger(\mathbf{k}), \hat{d}_\sigma(\mathbf{k})\}$, where $\mathbf{k}$ is the plane wave wavevector expanded around the CBM, and $\sigma$ denotes spin. The $e$-$e$ Coulomb interactions within substrate (Eq. (5d)) is taken to be the long-range Coulomb interaction with the same double-gate screened form of $V_{\text{int}}(\mathbf{q})$. The coupling between graphene and substrate is only via the long-range Coulomb potential, which is captured by Eq. (5e). The prefactor $e^2 \exp(-|\mathbf{q}| d)/(2\epsilon_0 \epsilon_r |\mathbf{q}|)$ in front of the field operators in Eq. (5e) is nothing but the 2D Fourier transform of 3D Coulomb potential. Interlayer hoppings can be neglected given that the interlayer distance $d \gtrsim 5$ Å in such heterostructures (e.g., $d \approx 7$ Å in graphene-CrOCl heterostructure from first principles calculations), thus the exponentially decaying interlayer hopping amplitude is much weaker than the power-law-decaying interlayer Coulomb interaction. This is further confirmed by directly calculating the orbital projected band structures of a commensurate supercell of CrOCl-graphene heterostructure based on density functional theory. It turns out that the Dirac cone in such heterostructure supercell stems almost 100% from carbon $p_z$ orbitals of graphene (see Supplementary Note 7 of Supplementary Information), which clearly indicates the absence of interlayer hybridization (hopping).

We use distinct letters to denote the ladder operators for electrons in graphene ($\hat{c}, \hat{c}^\dagger$) and substrate ($\hat{d}, \hat{d}^\dagger$). This implies in a

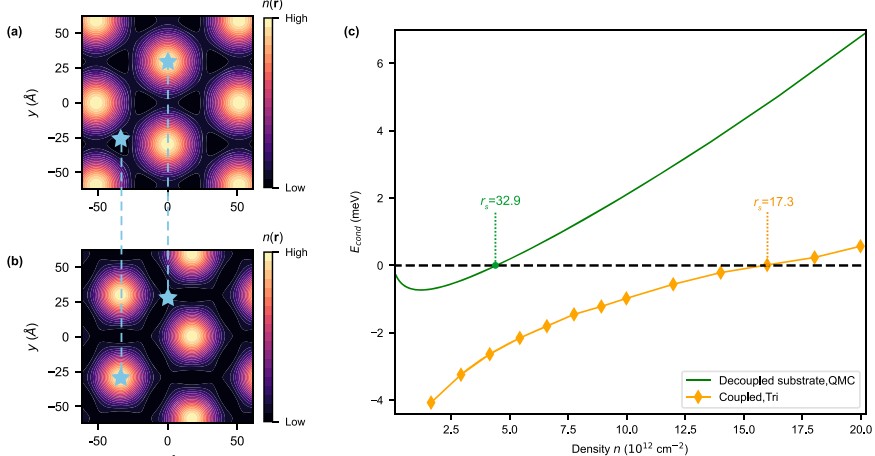

**Fig. 4 | Stabilizing effect of electronic crystal by interlocking of the charge modulations in coupled graphene-insulator heterostructures.** Charge density modulations $n(\mathbf{r})$ in real space after minimizing interlayer Coulomb interactions for **a** gapped Dirac state in graphene, and **b** electronic-crystal state in the substrate, which forms triangular superlattice. The blue stars annotate a maximum and a minimum of density in each state. The dashed lines connect two stars located at the same in-plane coordinate. **c** Condensation energy (per electron) of the electronic crystal state $E_{\mathrm{cond}}$ vs. the carrier density $n$ in the substrate. The green line represents the condensation energy of the decoupled system using the quantum Monte Carlo data, and the orange lines shows that of the coupled bilayer system, which is significantly stabilized by interlayer Coulomb coupling. The critical $r_s$ for Wigner-crystal transitions are also indicated in the coupled and decoupled cases, respectively.

notational manner the approximation of distinguishable electrons. In other words, the many-body wavefunction of the coupled bilayer system (denoted as $|\Psi\rangle$) can be written a separable fashion, namely a direct product of graphene's and substrate's part, i.e.,

$$|\Psi\rangle = |\Psi\rangle_c \otimes |\Psi\rangle_d \qquad (6)$$

In a mean-field treatment, the corresponding many-body wavefunction would thus be a direct product of two Slater determinants, $|\Psi\rangle_c$ and $|\Psi\rangle_d$ for the graphene layer and the substrate layer, respectively. This is reminiscent of the Born-Oppenheimer approximation for electrons and ions. Technically, this means that order parameters $\sim \langle \hat{c}^\dagger \hat{d} \rangle (\langle \hat{d}^\dagger \hat{c} \rangle)$ are not allowed in our treatment. A finite value of $\langle \hat{c}^\dagger \hat{d} \rangle (\langle \hat{d}^\dagger \hat{c} \rangle)$ suggests the emergence of another phase, an interlayer excitonic condensate in such coupled bilayer system. However, we note that such interlayer exciton has to be driven by intervalley Coulomb scattering between the $K/K'$ valley of graphene and (presumably) $\Gamma$ valley of substrate's electrons, with the amplitude $\sim e^2 \exp(-|\mathbf{K}|d)/(2\epsilon_0\epsilon_r|\mathbf{K}|)$ being several orders of magnitudes smaller than the intravalley one in our problem. Thus, it is completely legitimate to neglect the interlayer particle-hole exchange in our problem, and the separable wavefunction ansatz Eq. (6) is very well justified. Then, we solve the full interacting Hamiltonian Eqs. (5) under the separable wavefunction ansatz Eq. (6), and the workflow is presented in "Methods" section. Nevertheless, the interlayer excitonic insulator state consisted of Dirac electrons (holes) and quadratically dispersive holes (electrons) is possible in valley-matched graphene-insulator heterostructures, such as those consisted of graphene and transition metal dichalcogenides with the band extrema at $K$ and $K'$ points. We leave this for future study.

To explore how the interlayer Coulomb coupling would affect the electronic crystal state of the substrate, we first consider the situation as a reference that the substrate is decoupled from graphene. The energy difference between the spin polarized EC state and Fermi-liquid (FL) state (condensation energy) as a function of the carrier density $n$ is given by quantum Monte Carlo calculations in refs. 53,54, as shown by the green line in Fig. 4c, where an effective mass $m^* = 1.3$, a background dielectric constant $\epsilon_r = 4$, and valley degeneracy of 2 are considered in order to mimic the conduction band minimum of CrOCl. The condensation energy reaches zero when $n \approx 4.5 \times 10^{12}\,\mathrm{cm}^{-2}$ (corresponding to critical Wigner-Seitz radius $r_s^* \approx 32.9$), suggesting the transition from the EC to the FL state. More details are given in "Methods" section.

We further include the interlayer Coulomb coupling between the substrate and graphene (setting the chemical potential at the CNP of graphene), which can be treated using perturbation theory given that the interlayer Coulomb energy is always much smaller than the sum of the intralayer Coulomb energy and kinetic energy within the relevant parameter regime (see Supplementary Figure 12). Specifically, with the separable wavefunction ansatz (Eq. (6)), the ground-state charge densities for the graphene layer and the EC layer are separately obtained from unrestricted Hartree-Fock calculations, which are further used to estimate the interlayer Coulomb energy. More details about the perturbative treatment of interlayer Coulomb interactions are presented in Supplementary Note 6 of Supplementary Information.

We find that the condensation energy (per electron) of the EC is substantially enhanced in amplitude after including the interlayer interactions, as shown by the orange diamonds in Fig. 4. As a result, the EC-FL transition is postponed to a much higher density $n \approx 16 \times 10^{12}\,\mathrm{cm}^{-2}$ (corresponding to critical Wigner-Seitz radius $r_s^* \approx 17.3$). This is because the energy of the coupled bilayer can be further lowered by pinning the charge centers (marked as light blue stars in Fig. 4a, b) of the two layers in an anti-phase interlocked pattern, in order to optimize the repulsive interlayer Coulomb energy. The extra energy gain from such interlocking of charge centers compensates the energy cost of the EC state when $n \gtrsim 4.5 \times 10^{12}\,\mathrm{cm}^{-2}$, thus substantially stabilizes the EC state.

On the one hand, since the condensation energy of the free 2D electron gas in the decoupled substrate is estimated using the model that accurately fits to quantum Monte Carlo data[53], the estimate of the critical density for the decoupled substrate is expected to be accurate. On the other hand, in the case of substrate coupled with graphene layer, although the interlayer Coulomb energy is estimated with Hartree-Fock approximation, the qualitative conclusion (that the EC state gets stabilized by a cooperative interlayer Coulomb coupling) is expected to be valid even in a beyond-mean-field treatment. This is because under the separable wavefunction ansatz, the interlayer Coulomb energy in the EC state is always negative (compared to that of FL state) under an optimal choice of relative charge centers, which thus

always stabilizes the EC state even if the intralayer interactions are treated using beyond-mean-field approaches.

We note that the stabilizing effect of EC is not unique to band-aligned graphene-insulator heterostructures considered in this work. In principle, it only requires the presence of another state exhibiting non-uniform charge distribution atop of the EC, such that the inter-layer Coulomb energy gain would compensate for any energy cost of the long-wavelength charge modulations in the two layers. For example, remarkably robust EC state has been observed in a bilayer system consisting of two monolayer $MoSe_2$ separated by hexagonal boron nitride[55], which was also argued to be stabilized by the interlocking of the EC states in the two layers.

### Materials realization

The scenario discussed above is not only closely related to CrOCl-graphene and $CrI_3$-graphene heterostructures[19,21], but can also be extended to various band-aligned graphene-insulator hetero-structures. As along as the conduction band minimum (CBM) or valence band maximum (VBM) of the substrate is energetically close to the Dirac points of graphene, charges could be easily transferred between graphene and the substrate's surface by gate voltages. Fur-thermore, it is more likely to form long-wavelength ordered state at the surface of the substrate (with slight carrier doping) if the material has large effective masses at the CBM or VBM. Meanwhile, an insulator with relatively small dielectric constant would have weaker screening effects to $e$-$e$ interactions, which also favors long-wavelength ordered state at small carrier doping.

Following these guiding principles, we have performed high-throughput first principles calculations based on density func-tional theory for various insulating van der Waals materials. Eventually, we find twelve suitable candidate materials (including CrOCl and $CrI_3$), whose CBM and VBM energy positions, dielectric constants ($\epsilon_r$), effective masses at the band edges, and the corre-sponding Wigner-Seitz radii ($r_s$) are listed in Table 1. Clearly, the Wigner-Seitz radii of these materials at the band edges (estimated under slight doping concentration $n = 10^{12}$ cm$^{-2}$) are all above the threshold of forming a Wigner-crystal state ($r_s \gtrsim 31$)[53]. In addition, the energy bands of these insulating substrate materials can be easily shifted using vertical displacement fields (see Supplemen-tary Note 7 in Supplementary Information), such that charge transfer between graphene and the substrate can be controlled by non-disruptive gate voltages. We have also considered hetero-structures consisted of graphene and TMDs. Besides trilayer (or thicker) $WS_2$ as already listed in Table 1, we further nominate $WSe_2$ (trilayer or thicker), $MoSe_2$ (bilayer or thicker), and $MoTe_2$ (bilayer or thicker) as possible candidate substrates to realize the effects discussed above. More details are given in Supplementary Note 7 of Supplementary Information.

## Discussion

In summary, we have studied the synergistic correlated electronic states emerging from coupled graphene-insulator heterostructures with gate-tunable band alignment. Based on comprehensive theore-tical studies, we have shown that the gate-tunable carrier doping may yield a long-wavelength electronic crystal at the surface of the sub-strate driven by $e$-$e$ interactions within the substrate, which in turn exerts a superlattice Coulomb potential to the Dirac electrons in graphene layer. This would substantially change the low-energy spectrum of graphene, where a gapped Dirac state concomitant with drastically enhanced Fermi velocity would emerge as $e$-$e$ interaction effects. These theoretical results are quantitatively supported by our transport measurements in graphene-CrOCl heterostructure. Besides, the Dirac subbands in graphene can be endowed with non-trivial topological properties by virtue of the interlayer Coulomb coupling with the long-wavelength electronic crystal underneath.

Reciprocally, the electronic crystal in the substrate can be sub-stantially stabilized by virtue of a cooperative interlayer Coulomb coupling with the gapped Dirac state of graphene. We have further performed high-throughput first principles calculations, and sug-gested a number of promising insulating materials as candidate substrates for graphene to realize such effects.

However, the understanding of such coupled bilayer corre-lated electronic systems is still at a preliminary stage, and the study is far from being complete. First, the long-wavelength electronic crystal cannot be the only possible candidate ground state. Other correlated states such as magnetic or even super-conducting states may also occur in the charge doped insulating substrate, e.g., in the case of high-temperature cuprate superconductor[15,16] and monolayer 1T'-$WTe_2$[17]. This may give rise to diverse quantum states of matter in graphene due to interfacial proximity couplings with Dirac fermions. Moreover, so far we have only considered the ground state properties of such coupled bilayer correlated electronic systems. What is more intriguing is the collective excitations of the electronic crystal and their cou-plings with Dirac electrons in graphene. Around the quantum melting point of the electronic crystal, strong quantum fluctua-tions would be coupled with Dirac fermions with graphene via interlayer Coulomb interactions, which may give rise to unique quantum critical properties. Therefore, our work may stimulate further exploration of the intriguing physics in such a platform for correlated and topological electrons.

## Methods

### Hartree-Fock approximations assisted by renormalization group approach

When graphene is coupled to a superlattice potential, the Cou-lomb interactions are suitably expressed in the subband eigen-function basis, on which we have performed the Hartree-Fock calculations. Since interaction effects are most prominent around the CNP, we project the Coulomb interactions onto only a low-energy window including three valence and three conduction subbands that are closest to the Dirac point per valley per spin. We use a mesh of $18 \times 18$ **k**-points to sample the mini Brillouin zone of the superlattice.

To incorporate the influences of Coulomb interactions from the high-energy remote bands, we rescale the Fermi velocity within the low-energy window of the effective Hamiltonian using Eq. (4). The other parameters of the non-interacting effective Hamiltonian are unchanged under the RG treatment since their corrections are of higher order, thus can be neglected. In other words, we find the fol-lowing RG equations for Fermi velocity $v_F$ and leading superlattice potential $U_d$ with respect to energy cutoff $E_c$

$$\frac{d v_F}{d \log E_c} = -\frac{e^2}{16\pi\epsilon_0\epsilon_r}, \tag{7}$$

$$\frac{d U_d(\mathbf{Q})}{d \log E_c} = 0. \tag{8}$$

The detailed derivations of the RG equations are presented in Sup-plementary Note 3 of Supplementary Information.

We also neglect on-site Hubbard interactions and intervalley coupling in $e$-$e$ Coulomb interactions, which turn out to be one or two order(s) of magnitude weaker than the dominant intravalley long-range Coulomb interactions in such graphene-based superlattice systems[56]. To model the screening effects to the $e$-$e$ Coulomb inter-actions from the dielectric environment, we introduce the double gate

screening form of $V_{\text{int}}$, whose Fourier transform is expressed as

$$V_{\text{int}}(\mathbf{q}) = \frac{e^2 \tanh(qd_s)}{2\Omega_0 \epsilon_r \epsilon_0 q}, \qquad (9)$$

where $\Omega_0$ is the area of the superlattice's primitive cell, $\epsilon_r$ is a background dielectric constant and the thickness between two gates is $d_s = 400$ Å. Then, we initialize the Hartree-Fock loop with the initial conditions in the form of various different order parameters and obtain the converged ground state self-consistently (see Supplementary Note 4 of Supplementary Information).

When we consider electrons in graphene and substrate on equal footing in Eqs. (5), the routine of Hartree-Fock calculations is exactly the same. However, we need to first consider solely the substrate side. After performing unrestricted Hartree-Fock calculations, we use the ground-state charge density of EC in the substrate as input for constructing the superlattice potential. Explicitly, we need to replace Eq. (2) by

$$U_d(\mathbf{Q}) = \frac{e^2}{2\epsilon_0 \, \epsilon_r \, \Omega_0} \frac{e^{-|\mathbf{Q}|d} \, \rho_d(\mathbf{Q})}{|\mathbf{Q}|}. \qquad (10)$$

where $\rho_d(\mathbf{Q})$ is the Fourier component of the charge density in the substrate. More details can be found in Supplementary Note 6 of Supplementary Information.

**Workflow to solve the coupled bilayer Hamiltonian Eqs. (5)**
We solve the Hamiltonian of the coupled bilayer system described by Eqs. (5) in the following workflow:

- First, we start our calculations by considering solely the substrate Hamiltonian Eqs. (5b) and (5d). We considered the case of triangular superlattice, which is the actual ground state for the EC of 2D electron gas. In particular, the total energy of the triangular EC can described by a fitting model given in ref. 53:

$$E_{\text{WC}} = \frac{c_1}{r_s} + \frac{c_{3/2}}{r_s^{3/2}} + \frac{c_2}{r_s^2} + \frac{c_{5/2}}{r_s^{5/2}} + \frac{c_3}{r_s^3} \qquad (11)$$

where $c_1 = -1.106103$, $c_{3/2} = 0.814$, $c_2 = 0.113743$, $c_{5/2} = -1.184994$, and $c_3 = 3.097610$. These parameters are determined by fitting to quantum Monte Carlo data. The total energy for the Fermi-liquid state of 2D electron gas is given by the following model[54]:

$$E_{\text{FL}} = E_{\text{FL}}^{\text{HF}} + E_{\text{FL}}^c \qquad (12a)$$

$$E_{\text{FL}}^{\text{HF}} = \frac{1}{2r_s^2} - \frac{4\sqrt{2}}{3\pi r_s} \qquad (12b)$$

$$E_{\text{FL}}^c = a_0 \left\{ 1 + Ax^2 \left[ B \ln \frac{x+a_1}{x} + C \ln \frac{\sqrt{x^2 + 2a_2 x + a_3}}{x} + D \left( \arctan \frac{x+a_2}{\sqrt{a_3 - a_2^2}} - \frac{\pi}{2} \right) \right] \right\} \qquad (12c)$$

where $x = \sqrt{r_s}$ and

$$A = \frac{2(a_1 + 2a_2)}{2a_1 a_2 - a_3 - a_1^2} \qquad (13a)$$

$$B = \frac{1}{a_1} - \frac{1}{a_1 + 2a_2} \qquad (13b)$$

$$C = \frac{a_1 - 2a_2}{a_3} + \frac{1}{a_1 + 2a_2} \qquad (13c)$$

$$D = \frac{F - a_2 C}{\sqrt{a_3 - a_2^2}} \qquad (13d)$$

$$F = 1 + (2a_2 - a_1)\left( \frac{1}{a_1 + 2a_2} - \frac{2a_2}{a_3} \right) \qquad (13e)$$

with with $a_0 = -0.1925$, $a_1 = 7.3218$, $a_2 = 0.16008$, and $a_3 = 3.1698$. These parameters for the FL state are also determined by fitting to quantum Monte Carlo data[54]. The energies are given in Hartree atomic units. Then, one can extract the condensation energy for the isolated 2D electron gas in the substrate $E_{\text{WC}} - E_{\text{FL}}$, with the accuracy comparable to quantum Monte Carlo calculations.

- Second, with the help of the separable wavefunction ansatz Eq. (6), we further calculate the ground-state charge density of the EC state in the substrate under Hartree-Fock approximations. Although the Wigner crystal condensation energy would be significantly overestimated with such mean-field approximation, the ground-state charge density can still be properly described by the unrestricted Hartree-Fock treatment[57]. Then, one can integrate out the charge degrees of freedom of the substrate so that the charge density modulation characterized by the Fourier components of the charge density $\{\rho_d(\mathbf{Q})\}$ ($\mathbf{Q}$ denotes the reciprocal vector of the superlattice) can be used as an input for the superlattice potential $U_d(\mathbf{Q})$, as shown in Eq. (10). Compared to Eq. (2), this superlattice potential is more realistic and self-contained in our model. Equation (10) would be recovered to Eq. (2) by setting $\rho_d(\mathbf{Q}) = 2$ for any reciprocal vector $\mathbf{Q}$, which is equivalent to say that two (spin degenerate) charges per primitive supercell are localized in real space in a Dirac-$\delta$-function form.

- Third, we perform RG-assisted unrestricted HF calculations for the interacting Dirac electrons in graphene as explained in "Methods". If the chemical potential is at the CNP of graphene, a gap opening will be triggered by $e$-$e$ interactions within the graphene layer as discussed previously.

- From the above procedures, we would separately obtain converged HF ground states, $|\Psi\rangle_d$ for the substrate, and $|\Psi\rangle_c$ for graphene, respectively. From the ground-state wavefunctions $|\Psi\rangle_d$ and $|\Psi\rangle_c$, one can extract the corresponding ground-state charge density modulations $\{\rho_d(\mathbf{Q})\}$ and $\{\rho_c(\mathbf{Q})\}$, based on which the interlayer Coulomb energy (the expectation value of Eq. (5e)) can be calculated. More details are given in Supplementary Note 6 of Supplementary Information.

However, the ground states are obtained so far by minimizing (mostly) the intralayer parts of the full Hamiltonian, the interlayer Coulomb interaction Eq. (5e) is not optimized yet. We note that the intralayer kinetic energy and intralayer Coulomb interaction energy for both graphene and the substrate are unchanged under constant lateral shifts of the charge centers, thus the ground state $|\Psi\rangle_d \otimes |\Psi\rangle_c$ obtained so far is massively degenerate up to global and relative shifts of the bilayer charge centers. Such degeneracy would be partially lifted by the interlayer Coulomb energy $\langle H_{\text{gr-sub}} \rangle$. Obviously, $\langle H_{\text{gr-sub}} \rangle$ is invariant under the global shift of the charge centers of the bilayer system, but it varies with respect to a relative charge-center shift. Therefore, by virtue of perturbation theory, optimizing the interlayer Coulomb energy amounts to find the optimal relative shift vector between the charge centers of the two layers within the degenerate ground-state manifold obtained in the previous procedures. Such perturbative treatment of $H_{\text{gr-sub}}$ is justified given that the interlayer Coulomb energy is always weaker than the sum of

the kinetic energy and the intralayer Coulomb energy within relevant parameter regime, as shown in Supplementary Figure 12. For example, the interlayer Coulomb energy ~ 20 meV for typical parameters $L_s = 50$ Å and $\epsilon_r = 4$, while the intralayer Coulomb energy ~ 60 meV. More details for the perturbative calculation of interlayer Coulomb energy can be found in Supplementary Note 6 of Supplementary Information.

- Finally, we gather all the contributions from Eq. (5) to find out the total energy of the coupled bilayer system staying in a gapped Dirac state (at the CNP) for graphene and a long-wavelength EC state for the substrate. By comparing it with that of a non-interacting Dirac state for graphene and a 2D Fermi-liquid state for the substrate, we can then find out if the gapped graphene interplays with the long-wavelength charge-ordered substrate in a cooperative or competitive manner.

It turns out that the bilayer system tends to cooperate with each other such that both the gapped Dirac state (at the CNP) of graphene and the long-wavelength charge ordered state in the substrate are substantially stabilized by the interlayer Coulomb coupling. The results are presented in Figs. 2 and 4 of the main text.

### Density functional theory calculations

The first principles calculations are performed with the projector augmented-wave method within the density functional theory[58], as implemented in the Vienna ab initio simulation package software[59]. The crystal structure is fully optimized until the energy difference between two successive steps is smaller than $10^{-6}$ eV and the Hellmann-Feynman force on each atom is less than 0.01 eV · Å. The generalized gradient approximation by Perdew, Burke, and Ernzerhof is taken as the exchange-correlation potential[60]. As Cr is a transition metal element with localized $3d$ orbitals, we use the on-site Hubbard parameter $U = 5.48$ eV for the Cr $3d$ orbitals in the CrOCl bilayer and $U = 3$ eV for Cr $3d$ orbitals in the CrI$_3$ bilayer. The so-called fully localized limit of the spin-polarized GGA+U functional is adopted as suggested by Liechtenstein and coworkers[61], and the non-spherical contributions from the gradient corrections are taken into consideration. The "DFT+D2" type of vdW correction has been adopted for all multilayer calculations to properly describe the interlayer interactions[62].

Our high-throughput filtering of the proper insulating substrate materials for graphene starts from the 2D materials computational database[63]. We only focus at those with bulk van der Waals structures which have been previously synthesized in laboratory. This ensures that it is experimentally feasible to exfoliate few layers from their bulk sample and then stack them on graphene to form heterostructures.

### Experimental measurements of the gaps in graphene-CrOCl heterostructure

By designing a dual-gated structure, we used few-layered CrOCl as a bottom dielectric while few-layered hexagonal boron nitride (h-BN) was served as top gate dielectric. The top and bottom gate voltages can then be converted into doping and displacement fields for further data analysis. Graphene, h-BN, and CrOCl flakes are mechanically exfoliated from high-quality bulk crystals. The vertical assembly of few-layered hBN, monolayer graphene, and few-layered CrOCl were made using the polymer-assisted dry-transfer method. Electron beam lithography was done using a Zeiss Sigma 300 SEM with a Raith Elphy Quantum graphic writer. Top and bottom gates as well as contacting electrodes were fabricated with an e-beam evaporator, with typical thicknesses of Ti/Au ~ 5/50 nm. Electrical transport measurements of the devices were performed using an Oxford TeslaTron 1.5 K system. Gate voltages on the as-prepared multi-terminal devices were fed by a Keithley 2400 source meter. Channel resistances were recorded in 4-probe configurations using low frequency (13.33 Hz) lock-in

technique with Stanford SR830 amplifiers. The gate dependencies of channel resistances were measured at various temperatures for the extraction of thermal gaps. More details about the device configuration, measurement set-up, and sample quality can be found in Supplementary Note 8 of Supplementary Information.

## Data availability
The data that support the findings of this study are available at https://figshare.com/projects/MonoGr-CrOCl/174702.

## Code availability
The codes that support this study are available from the corresponding author upon request.

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

## Acknowledgements

We would like to thank Jian Kang and Jinhai Mao for valuable discus-sions, and to thank Hanwen Wang for the help in making the plots. X.L., S.Z., Z.G., and J.L. acknowledge support from the National Key R & D program of China (grant No. 2020YFA0309601), the National Natural Science Foundation of China (grant No. 12174257), and the start-up grant of ShanghaiTech University. Y.W., X.G., K.Y., and Z.H. acknowledge support from the National Key R & D program of China (grant No.

2022YFA1203903) and National Natural Science Foundation of China (grant No. 92265203, 11974357).

## Author contributions

J.L. conceived the idea, constructed the theoretical model, and supervised the project. X.L., S.Z., Z.G., and J.L. performed theoretical calculations. Y.W., X.G., K.Y., Y.G., Y.Y., and Z.H. performed transport measurements. X.L., Z.G., Z.H., and J.L. analyzed the data. X.L. and J.L. wrote the manuscript with inputs from all authors.

## Competing interests

The authors declare no competing interests.
