## [Peer review file · Nature Communications]

REVIEWER COMMENTS

Reviewer #1 (Remarks to the Author):

Motivated by recent experiments, the authors study interaction effects of coupled graphene-insulator heterostructures. The authors first study single particle and mean field band structures of graphene in a periodic potential and find that the Fermi velocity of Dirac electrons can be significantly enhanced by interactions. Bands with nontrivial Chern number are also discovered by tuning the anisotropy of the periodic potential. The authors then study the interplay between an electron lattice and graphene and find that the graphene can enhance the stability of the electron lattice.

The manuscript may help experimentalists stabilize electron lattices and has the potential to be published in Nature Communications. However, there are several issues that must be addressed:

1. The authors show a significant enhancement of Fermi velocity by interactions in Fig. 2b. However, the authors then claim that gap opening at CNP leads to two nearly degenerate insulating states and one of them is valley polarized. What does it mean exactly to be valley polarized? If it means that only one valley is populated, how can the system overcome such large band width and be valley polarized?
2. For CrOCl and CrI₃, the band structures cannot be simply described by a single valley. In addition, both materials are either magnetic or antiferromagnetic. What would these complications affect the results of the paper?
3. Electron lattices are shown to be more stable by forming graphene-insulator heterostructure. What is the special role of graphene in this finding? Will any 2D metal-insulator heterostructure also stabilize electron lattices?

Minor points:

1. In line 121, there is a missing reference to a session.
2. The y axis of Fig. 2e has typo.

Reviewer #2 (Remarks to the Author):

"Synergistic correlated states and nontrivial topology in coupled graphene-insulator heterostructures" is a well written manuscript, including a complete and detailed supplementary material which answers most of my questions and doubts during my reading.

This manuscript, as an elaborate theoretical part of experiment Ref.[19] (with common authors), carefully study the correlated states in graphene when it is in proximity to a superlattice EC. Unlike in the so-called moire superlattices (such as TBG), graphene and EC are only coupled through Coulomb interactions. On the single-particle level, the authors first construct a low-energy effective model of Dirac fermions in a long-wavelength charge order of EC. Such a long-wavelength charge order folds graphene's original BZ, quenching the kinetic energy of Dirac fermions and making Coulomb interactions crucial. Then using the Hartree-Fock approach, the authors find that strong Coulomb interactions gap the Dirac cone and enhance the Fermi velocity near CNP. Moreover nontrivial topological bands can be realized by fine tuning superlattice's constant and anisotropy.

To be honest, the work reported in this manuscript is worthwhile from the theoretical understanding point of view and is also applicable to guide future experiments. First, theoretical assumptions of the low-energy effective model Eq.(1) are justified by DFT calculations; Second, the authors explore a

variety of substrate candidates using DFT; Third, their theoretical model is verified, to some extent, by experiments (for example the calculated gap Fig.2d agrees very well with experimental measurement in Fig.2e). Even though I'm not sure about this work's originality, but I think it is worthwhile to publish based on significances above.

But I still have several questions and find one typo in the manuscript.

1. Throughout the manuscript, the geometry of the superlattice EC is assumed to be rectangular or triangular. For example in line 63, a rectangular superlattice EC with a tunable anisotropy is used. The authors also comment in the SM S1 that the superlattice geometry makes no qualitative differences to the physics reported in this paper. But an intuitive understanding of this independence of the underlying superlattice geometry is lacking.

2. In DFT calculations, the atomic lattices of the insulating substrate must be commensurate with graphene carbon lattices. However, the requirement of this commensurability is lifted in the continuum model Eq.(1). If I understand correctly, the commensurability is not required and not important if the superlattice wavelength is much larger than carbon atomic lattice constant, and this condition is always satisfied throughout the paper.

3. There's a tiny energy shift in red dashed lines in fig.2(c) compared to that in fig.2(b). The single-particle band structure should not depend on the filling factor.

4. There's missing text In line 121, 'which will be discussed in detail in Sec. .' .'

Reviewer #3 (Remarks to the Author):

The authors studied the interaction between a single-layer graphene and an insulator substrate. The electron crystal formed in the insulating layer provided a superlattice potential for graphene, resulting in the reduction of Fermi velocity. Furthermore, the authors considered the electron-electron interactions in graphene which led to gap opening at the Fermi level, and the non-trivial topology along with. While the numerical calculations and the theoretical analysis may provide better insight into the graphene-related research, the relation between the computation results and the conclusions is not decisive, and the significance of the research may not be high enough for publication in Nature Communication. Particularly, there are several issues that needs clearer illustration:

0. Part of the results have been reported by the authors' previous paper in Nat. Nano.

1. The authors assumed from the very beginning that the insulating layer with rotationally symmetric 2DEG dispersion will form electron crystal phase with rectangular lattice. They did Hartree-Fock calculations to show that the total energy is lower than electron liquid phase. However, electron crystal is a strongly correlated system, and Hartree-Fock calculations are not enough to describe 1. how the Wigner crystal melts, and 2. how graphene layer stabilizes the crystalline phase. The single-particle plane wave basis is not an appropriate starting point. Furthermore, rather than assuming the rectangular shape of the lattice without reasoning, the authors should compare the ground state energy among different real space configurations, triangular, hexagonal, rectangular, etc. Still, the estimation of condensation energy requires careful treatment of many-body correlations, especially when the authors are trying to give quantitative estimate ('extrapolation' in FIG. 4) of the critical density for phase transition to the electron liquid.

2. If there is charge transfer between the layers, why can one still treat the electrons in two layers separately, and consider the substrate layer simply as Dirac-delta potentials?

3. Line 152. What is special about these 'hot spots' that contribute to large Berry curvature? Is there a

way to identify their nature and how it changes with L_s and r ? The origin of new topological properties is not demonstrated clearly.

4. In supplement S1, 'The sum over Q in Eq. (S23) stops at the limit $|n_x| + |n_y| \leq 2$ '. What is the justification of not considering larger number of folded zones?

5. The authors show in S9 that with electrical transport measurement of the graphene-CrOCl heterostructure, gap opening at CNP can be observed from the longitudinal resistivity peak. They determined the gap size by fitting temperature-dependence of resistivity with $\exp(-\Delta/2k_B T)$. The link between experiment and theory is poor. Firstly, the authors should show that the sample quality is good enough, so that the resistivity peak is indeed from a gapped phase. At least images of the experimental configuration and the measurement setup should be provided. Secondly, there is no evidence of Wigner crystal formation, and the authors did not rule out any other possible factors that may lead to gap opening.

And some grammatical and arrangement issues:

1. Organization can be better. Too much repetition of long paragraphs between main texts and supplement.
2. Line 121 'Sec..'
3. Line 194-202 are lengthy. Please refine the words.
4. S2: "We also provide separately six videos in other Supplementary Information" which cannot be found.
5. Above (S65) 'less harsher'
6. Below Fig. S11 'given by sing the charge'
7. A few other obvious typos in the supplement such as 'from' rather than 'form'.

Dear Reviewers,

We would like to thank the three reviewers for their insightful questions and comments (Q&Cs), which truly help a lot in improving the quality of our manuscript. Following the reviewers' Q&Cs, we have performed extensive additional calculations, and have significantly revised the manuscript.

In this response, we will first reply to the Q&Cs from the three reviewers point by point, then we will make a list of the significant changes made in the resubmitted manuscript at the end. For clarity's sake, the Q&Cs from the reviewers are marked in blue.

Reviewer #1 (Remarks to the Authors):

Motivated by recent experiments, the authors study interaction effects of coupled graphene-insulator heterostructures. The authors first study single particle and mean field band structures of graphene in a periodic potential and find that the Fermi velocity of Dirac electrons can be significantly enhanced by interactions. Bands with nontrivial Chern number are also discovered by tuning the anisotropy of the periodic potential. The authors then study the interplay between an electron lattice and graphene and find that the graphene can enhance the stability of the electron lattice.

The manuscript may help experimentalists stabilize electron lattices and has the potential to be published in Nature Communications. However, there are several issues that must be addressed:

Reply: We thank the reviewer for the concise and clear summary of our work. The reviewer has captured all the key points of our work. We are also grateful to the reviewer for his approval of the impact of our work. Concerning the specific Q&Cs posed by the reviewer, we will answer them point-to-point.

Q&C 1. The authors show a significant enhancement of Fermi velocity by interactions in Fig. 2b. However, the authors then claim that gap opening at CNP leads to two nearly degenerate insulating states and one of them is valley polarized. What does it mean exactly to be valley polarized? If it means that only one valley is populated, how can the system overcome such large band width and be valley polarized?

Reply: We thank the referee for asking this crucial question. First, we apologize our misuse of the term “valley-polarized state”, which refers to the interacting ground state characterized by τ_z in most of literatures (here $\tau = (\tau_x, \tau_y, \tau_z)$ refer to Pauli matrices defined in valley space). In the legacy version of the paper, this referred by abuse of language to the gapped state characterized by $\tau_z\sigma_z$, where σ_z is the third Pauli matrix defined in sublattice space. The $\tau_z\sigma_z$ -state is not topological (as the subbands are topologically trivial for the particular case considered in the Hartree-Fock calculations), and is quasi-degenerate to the sublattice polarized state characterized by order parameter σ_z . However, while including the intervalley Coulomb interaction (which is two orders of magnitudes weaker than the intravalley one) in the Hartree-Fock calculations, the σ_z -state is stabilized and becomes the unique interacting ground state.

This is clearly shown in Table R1. When only intravalley Coulomb interactions (gray figures in the parentheses) are included, the energy difference between $\tau_z\sigma_z$ and σ_z states is 8×10^{-9} eV, smaller than the preset convergence threshold (10^{-8} eV) for the self-consistent Hartree-Fock calculations. While if both inter-valley and intra-valley Coulomb interactions (black figures out of parentheses) are taken into account, the energy difference between $\tau_z\sigma_z$ and σ_z states rises up to $\sim 10^{-6}$ eV.

Dominant order parameter	Energy difference w.r.t. σ_z -state (eV)	Gap (eV)	Valley Chern number	
			VB	CB
σ_z	0 (0)	0.0159 (0.0156)	0	0
$\tau_z\sigma_z$	1.63738E-06 (8.13804E-09)	0.0153 (0.0156)	0	0

Table R1 Three possible interacting states for $L_s = 50 \text{ \AA}$, $\epsilon_r = 3.0$ in rectangular lattice $r = 1.2$. In the paratheses, we list the energy differences calculated without intervalley Coulomb interactions. Out of the parathesis, we give the results with both inter- and intra-valley Coulomb interactions.

In Fig. R1(a) we present Hartree-Fock band structures of the $\tau_z\sigma_z$ state. Since this state breaks the combined $C_{2z}T$ symmetry (T for time-reversal) which protects the degeneracy of Dirac point, the Dirac point at Γ_s is indeed gapped out as clearly shown in the zoom-in band structure Fig. R1(b). It is worthwhile to note that since the dominant order parameter is $\tau_z\sigma_z$ instead of τ_z , the net valley polarization vanishes in this state.

Fig. R1 Hartree-Fock band structures of the $\tau_z\sigma_z$ -state for $L_s = 50 \text{ \AA}$, $\epsilon_r = 3.0$ in rectangular lattice $r = 1.2$. In (a), we show the full bands; in (b), we show the details at Γ_s point.

In response to the referee's comment, we have rewritten the relevant part in the main text and Sec. S5 in Supplementary Material to better illustrate the paragraphs above. In particular, we attribute explicitly the characterizing order parameter to the states while discussing them in both the main text and Supplementary Material. In line 117 of the main text, we have the changed the sentence

"... leading to two nearly degenerate insulating states, one is sublattice polarized and the other valley polarized"

to the following:

"...leading to two nearly degenerate insulating states, one is σ_z -sublattice polarized and the

other is characterized by the order parameter $\tau_z\sigma_z$, where τ_z and σ_z denote the third Pauli matrix in the valley and sublattice space, respectively."

Q&C 2(a). For CrOCl and CrI₃, the band structures cannot be simply described by a single valley.

Reply: We thank the reviewer for the crucial comments. When estimating the Wigner-Seitz radii of the different substrates (Table I in the main text), the valley degeneracy has already been taken into account. However, when calculating the properties of graphene (including Coulomb potential from the substrate's Wigner crystal), we admit that we have only considered the case of single valley. This is because valley degeneracy is a material dependent property, and we would like to first capture the universal physics that is independent of materials' details. In reality, the conduction band minimum of both CrOCl and CrI₃ are two-fold degenerate (see Table I). After including the two-fold valley degeneracy of the substrate, we find that our theoretical results are still qualitatively consistent with those from single-valley calculations. Most saliently, after taking into account the twofold valley degeneracy, for CrOCl-graphene heterostructure, the calculated gaps of graphene fit even more precisely to the experimental measured one.

Specifically, the experimental data (Fig. 2(e) and Fig. S19(c)) shows that the gap decreases linearly with the total carrier density (n_{tot}) at least for $0.6 \times 10^{12} \text{ cm}^{-2} < n_{tot} < 3.5 \times 10^{12} \text{ cm}^{-2}$. When n_{tot} goes to zero, the gap is eventually closed as indicated by both theoretical calculations and experimental data. This is consistent with the intuition that if there is no carrier present in the substrate, the formation of electronic crystal is irrelevant and graphene is trivially deposited on an insulating substrate so that Dirac cone remains gapless. Although it is hard to experimentally access the gap at very low carrier density, we can still depict two possible situations for the evolution of gap as n_{tot} vanishes: the gap would vanish either abruptly as a first-order insulator-to-metal transition, or continuously. The two scenarios are schematically shown in Fig. R2(a) and (b), respectively. In the case of a continuous insulator-to-metal transition, the gap cannot always decrease linearly with n_{tot} following the slope in the $n_{tot} \sim 10^{12} \text{ cm}^{-2}$ regime; rather at low carrier densities $n_{tot} \sim 10^{11} \text{ cm}^{-2}$ the gap has to decrease more rapidly and eventually drops to zero continuously as shown in Fig. R2(b).

However, although the previously calculated gap vs. n_{tot} (Fig. 2(d)) indeed exhibits a linear relationship which is consistent with the experimental data in the $n_{tot} \sim 10^{12} \text{ cm}^{-2}$ regime, there is still some small discrepancy: the calculated gap vs. n_{tot} slope is larger than the experimentally measured one. As a result, our previous results in Fig. 2(d) of main text indicate that the calculated gap vs. n_{tot} exhibits an almost linear relationship for all carrier densities of interest ($0 < n_{tot} \leq 7 \times 10^{12} \text{ cm}^{-2}$), which is not fully consistent with either of the two scenarios shown in Fig. R2.

Fig. R2 Two possible scenarios for the gap closing as n_{tot} goes to 0: (a) first-order transition or (b) continuous transition. The dashed lines in (a) and (b) mark the region of n_{tot} within which the experimental data for the gaps are available.

Our new Hartree-Fock calculations for graphene including two-fold valley degeneracy of CrOCl substrate suggest that it is the second situation (Fig. R2(b)) that happens in graphene-CrOCl heterostructure. Specifically, at given superlattice constant L_s (with fixed anisotropy parameter $r=1.2$, same as that of CrOCl), the Coulomb potential from the Wigner crystal in the substrate is expressed as

$$U_d(\mathbf{Q}) = e^2 g_v \frac{n(\mathbf{Q}) e^{-|\mathbf{Q}|d}}{2\epsilon_0 \epsilon_r \Omega_0 |\mathbf{Q}|} \quad (\text{R1})$$

where $g_v=2$ is the valley degeneracy, $n(\mathbf{Q})$ is the Fourier transformed charge density per valley at superlattice's reciprocal vector \mathbf{Q} , ϵ_r is the background dielectric constant, and $d = 7 \text{ \AA}$ is the interlayer distance between CrOCl and graphene based on our DFT calculations. In the limit of δ -function like localized charge density, $n(\mathbf{Q}) = 2$ for all \mathbf{Q} (factor of 2 from spin degeneracy). When the filling factor of graphene is set to charge neutrality point, the total carrier density is then $n_{tot} = 2g_v/\Omega_0 = 4/\Omega_0$, where $\Omega_0 = rL_s^2$ is the area of the supercell. Using the superlattice potential in Eq. (R1), we further include e-e interactions within graphene layer and perform Hartree-Fock calculations as already explained in detail in the manuscript. The results are given in Fig. R3, where we use two dielectric constants of CrOCl ($\epsilon_r = 3.7$ and 4.0). At $n_{tot} > 10^{12} \text{ cm}^{-2}$, the theoretical calculations show a linear behavior of the gap vs. n_{tot} relationship, with the calculated slope being quantitatively the same as the experimental measured one. In particular, the calculated gaps with $\epsilon_r = 3.7$ are almost exactly the same as the experimental measured values. When n_{tot} approaches zero, the slope becomes steeper and the gap is eventually closed continuously, perfectly consistent with the second scenario in Fig. R2(b).

In summary, our results illuminate the general evolution behavior of gap while n_{tot} approaching zero: when n_{tot} is large ($\sim 10^{12} \text{ cm}^{-2}$), the gap decays linearly with density in a material-dependent decaying rate; when n_{tot} approaches 0, the gap would adjust its slope assuring that the gap is closed at zero carrier density. However, we would like to emphasize that the valley degeneracy of the substrate is a material dependent property.

The results presented in Fig. 2(d) of the main text (without including valley degeneracy) and in Fig. R3 are actually qualitatively consistent with each other: in both cases the gap show a linear dependence on n_{tot} in the $n_{tot} \sim 10^{12} \text{ cm}^{-2}$ regime, and continuously drops to zero as $n_{tot} \rightarrow 0$. The only quantitative difference is the gap vs. n_{tot} slope, which seems to be crucially dependent on the valley degeneracy.

In response to the reviewer's comments, we have added the following sentences in the main text (starting from line 131) to discuss effects of valley degeneracy:

"This is because in Eq. (2), the interlayer Coulomb potential only applies to the situation of a single valley to accommodate the charge carriers in the substrate. In reality, there may be additional valley degeneracy in the substrate, which is crucial for the evolution of gap for $n_{tot} \rightarrow 0$. Although the valley degeneracy of the substrate does not change our results qualitatively, the theoretically calculated gap vs. n_{tot} fits to the experimental data of CrOCl-graphene heterostructure more precisely at low density once including the two-fold valley degeneracy of CrOCl (see Table I). The details are given in Supplementary Material (Fig. S11) [22]."

We have also included Fig. R3 into in Sec. S5 of Supplementary Material as well as a new paragraph explaining the results on graphene-CrOCl heterostructure at the end of that section.

Again, we thank the reviewer's insightful comments, which truly help us to better understand the physics in this system.

Fig. R3 Comparison between the experimentally measured gap and the theoretically calculated gaps.

Q&C 2(b) In addition, both materials are either magnetic or antiferromagnetic. What would these complications affect the results of the paper?

Reply: The reviewer posed a very interesting question: what would happen if there is magnetic proximity couplings between a slightly charge doped magnetic substrate and

graphene.

On the one hand, if the ferromagnetism or antiferromagnetism in the substrate is unchanged by small amount of charge doping, the “trivial” magnetism of the substrate could still affect graphene via proximity coupling. Since the magnetism breaks time-reversal symmetry, Dirac cone is no longer protected by $C_{2z}T$ and thus gapped in the non-interacting level. According to previous studies (e.g., *Phys. Rev. Lett.* 112.116404), when graphene is proximity coupled to an antiferromagnetic insulator (such as BiFeO₃), the proximity-induced exchange field with Rashba spin-orbit coupling could make graphene a Chern insulator exhibiting quantum anomalous Hall effect. When graphene is placed on top of a ferromagnetic insulator such as Cr₂Ge₂Te₆ (*2D Mater.* 7 (2020) 015026), it was also theoretically suggested that the magnetic proximity effect may induce spin splitting in graphene.

On the other hand, when the magnetic substrate is slightly charge doped such that the Wigner-Seitz radius is above threshold to form a Wigner crystal, then a long-wavelength magnetic order commensurate with the Wigner crystal may be formed. The long-wavelength magnetic order may be associated with nontrivial magnetic textures such as skyrmions, vortices, etc., especially if Dzyaloshinskii–Moriya interaction is present. In such a situation, the magnetic texture may be transmitted to graphene’s layer via interfacial magnetic proximity couplings, which may generate topologically nontrivial flat bands on graphene’s side, and can potentially lead to diverse correlated and topological states. This would be an interesting topic by itself, which deserves further studies.

However, it turns out that all the effects discussed above are so far still on the theoretical level. Magnetic proximity effects have not been observed for CrOCl- and CrI₃-graphene heterostructures. For example, for our CrOCl-graphene device, no coercive field has been seen while scanning back and forth the magnetic field, and the Landau level degeneracy is still compatible with that of spin-valley degenerate Dirac cones (implying the absence of magnetic exchange field). Most saliently, the gap opening and the robust quantum Hall effect persist up to temperatures far above the Néel temperature of CrOCl (~14 K). For CrI₃-graphene device reported in *Nano Letters* 22, 8495 (2022), the Landau level degeneracy also obeys that of spin degenerate Dirac cones, which also indicate the absence of magnetic proximity effect from CrI₃ substrate. The authors also explicitly claimed in their abstract that for CrI₃-graphene heterostructure: “*Surprisingly, we are unable to detect a magnetic exchange field in the graphene ...*”. Therefore, for both CrOCl- and CrI₃-graphene heterostructures, it seems that magnetism in the insulating substrate do not affect the physics in the graphene layer. Instead, it is the charge transfer and the interlayer Coulomb coupling between the substrate and graphene that play the crucial role. For the case of CrOCl, this may be due to a large interlayer distance between graphene and (~7 Å) so that any short-range exchange effects are negligible.

In response to the reviewer’s question, we have added a new paragraph starting from line 156 of the main text in the resubmitted manuscript:

"Although it has been theoretically proposed that the magnetic proximity effect together with spin-orbit coupling could in principle give rise to topologically nontrivial states in graphene [46], it seems to be irrelevant to the graphene-insulator heterostructures considered in the present study. For example, in CrOCl-graphene device, no magnetic hysteresis has been observed in graphene, and the measured Landau level degeneracy is still compatible with that of spin-valley degenerate Dirac cones [19]. Most saliently, the gap opening and the robust quantum Hall effect persist up to temperatures far above the Neel temperature of CrOCl (~14 K) [19]. Similarly, the magnetic proximity coupling was also reported to be negligible for CrI₃-graphene heterostructure [21]. Therefore, compared to the power-law decaying interlayer Coulomb coupling, the exponentially decaying magnetic proximity coupling may not play an important role in such charge-transfer graphene-insulator heterostructures."

Q&C 3. Electron lattices are shown to be more stable by forming graphene-insulator heterostructure. What is the special role of graphene in this finding? Will any 2D metal-insulator heterostructure also stabilize electron lattices?

Reply: We thank the referee for asking this insightful question.

The referee is right that the mechanism could be more general. The stabilizing effect of Wigner crystal only requires the presence of another gapped state exhibiting non-uniform charge distribution atop of the Wigner crystal. For example, we just noticed that Wigner crystals have been observed in a bilayer system consisting of two monolayer MoSe₂ separated by hBN (*Nature volume 595, pages48–52 (2021)*). Wigner crystal in this system is remarkably robust and persists at temperature up to 40 K and at electron density up to $6 \times 10^{12} \text{ cm}^{-2}$ without help of magnetic field or moiré potential. The authors attribute the unexpected stability of Wigner crystal in the bilayer system to interlayer Coulomb potential, because two Wigner crystal lattices are interlocked. This shows the importance of interlayer Coulomb potential in the emergence of Wigner crystal, which might be overlooked for long time.

In our work, we reveal in a new system the crucial role that interlayer Coulomb potential would play. When we separately consider an electron-doped substrate (with carrier density exceeding the threshold to form a Wigner crystal) and charge neutral graphene, none of them would experiences metal-to-insulator transition because Fermi liquid state and gapless graphene are more energetically favorable, respectively for their own. However, when we couple the two layers via interlayer Coulomb potential, both of them could become gapped. This is because the energy gain by optimizing the relative charge centers between Wigner crystal and gapped graphene could overcome the energy cost, namely:

$$E_{\text{inter}} + E_{\text{WC}} + E_{\text{gapped,gr}} - E_{\text{FL}} - E_{\text{gapless,gr}} < 0 , \quad (\text{R2})$$

where E_{WC} and E_{FL} refer to the total energies of the Wigner-crystal and Fermi-liquid states of the charge-doped substrate, respectively. $E_{\text{gapped,gr}}$ and $E_{\text{gapless,gr}}$ denote the

total energies of the gapped and gapless graphene, respectively. E_{inter} denotes the interlayer Coulomb energy between the two gapped states. In general, graphene can be replaced by another layer that may undergo metal-to-insulator transition at proper fillings, and Eq. (R2) can be replaced by a more general condition

$$E_{\text{inter}} + E_{\text{WC}} + E_{\text{gapped}} - E_{\text{FL}} - E_{\text{gapless}} < 0 , \quad (\text{R3})$$

where E_{gapless} and E_{gapped} refer to total energies of the gapless and gapped states in the other layer. As long as Eq. (R3) is fulfilled for a bilayer heterostructure system, the Wigner crystal state would be stabilized

Comparing to other 2D metals, we expect that electronic crystal phases might be easier experimentally accessed in graphene-insulator heterostructures than in normal 2D metal-insulator ones. On the one hand, graphene can be in principle gapped by interactions (though not observed before but proved by quantum Monte Carlo simulations) at charge neutral point, which can be easily tuned using gate voltages. On the other hand, the low density of states at Dirac point induces a smaller screening effect. Nevertheless, we believe that such effect should be relevant in other 2D metal-insulator heterostructure as long as the 2D metal is susceptible to a charge order state triggered by e-e interactions or other possible mechanisms.

In response to the reviewer's question, we have added a new paragraph in main text starting from line 306 of the resubmitted manuscript:

"We note that the stabilizing effect of EC is not unique to band-aligned graphene-insulator heterostructures considered in this work. In general, it only requires the presence of another (tunable) gapped state exhibiting non-uniform charge distribution atop of the EC. For example, remarkably robust EC state has been observed in a bilayer system consisting of two monolayer MoSe₂ separated by hexagonal boron nitride [57], which was also argued to be stabilized by the interlocking of the EC states in the two layers."

Minor points:

- 1. In line 121, there is a missing reference to a session.**
- 2. The y axis of Fig. 2e has typo.**

Reply: We sincerely thank the reviewer for pointing out these typos and careless mistakes. All of them have been fixed in the resubmitted manuscript.

Reviewer #2 (Remarks to the Author):

"Synergistic correlated states and nontrivial topology in coupled graphene-insulator heterostructures" is a well written manuscript, including a complete and detailed supplementary material which answers most of my questions and doubts during my reading.

This manuscript, as an elaborate theoretical part of experiment Ref.[19] (with common authors), carefully study the correlated states in graphene when it is in proximity to a superlattice EC. Unlike in the so-called moire superlattices (such as TBG), graphene and EC are only coupled through Coulomb interactions. On the single-particle level, the authors first construct a low-energy effective model of Dirac fermions in a long-wavelength charge order of EC. Such a long-wavelength charge order folds graphene's original BZ, quenching the kinetic energy of Dirac fermions and making Coulomb interactions crucial. Then using the Hartree-Fock approach, the authors find that strong Coulomb interactions gap the Dirac cone and enhance the Fermi velocity near CNP. Moreover nontrivial topological bands can be realized by fine tuning superlattice's constant and anisotropy.

To be honest, the work reported in this manuscript is worthwhile from the theoretical understanding point of view and is also applicable to guide future experiments. First, theoretical assumptions of the low-energy effective model Eq.(1) are justified by DFT calculations; Second, the authors explore a variety of substrate candidates using DFT; Third, their theoretical model is verified, to some extent, by experiments (for example the calculated gap Fig.2d agrees very well with experimental measurement in Fig.2e). Even though I'm not sure about this work's originality, but I think it is worthwhile to publish based on significances above.

But I still have several questions and find one typo in the manuscript.

Reply: We are flattered that the reviewer found our manuscript to be "well written" and that the Supplementary Material helped the reviewer to better understand our work. We thank the reviewer for recognizing the significance of our manuscript. We also heartily appreciate the reviewer's recommendation on the publication of our manuscript in Nature Communications. Concerning the questions posed by the reviewer, we will answer them point-to-point.

Q&C 1. Throughout the manuscript, the geometry of the superlattice EC is assumed to be rectangular or triangular. For example, in line 63, a rectangular superlattice EC with a tunable anisotropy is used. The authors also comment in the SM S1 that the superlattice geometry makes no qualitative differences to the physics reported in this paper. But an intuitive understanding of this independence of the underlying superlattice geometry is lacking.

Reply: We thank the referee for asking this essential question.

The interacting physics in graphene is governed by the effective fine-structure constant α , which is proportional to Fermi velocity of Dirac cone and inversely proportional to the dielectric constant. In the presence of a superlattice Coulomb potential, the Fermi velocity is reduced. Treating the superlattice potential using second-order perturbation theory, in the most general case, the renormalized non-interacting effective Hamiltonian can be expressed as,

$$H_{\text{eff}}^0(\mathbf{k}) = \hbar v_F^0 \left(1 - \sum_{|\mathbf{Q}| \neq 0} \frac{|U_d(\mathbf{Q})|^2}{(\hbar v_F^0)^2 |\mathbf{Q}|^2} \right) \left(\mathbf{k} - \sum_{|\mathbf{Q}| \neq 0} \frac{|U_d(\mathbf{Q})|^2}{(\hbar v_F^0)^2 |\mathbf{Q}|^2} \left(\mathbf{k} - \frac{2\mathbf{k} \cdot \mathbf{Q}}{|\mathbf{Q}|^2} \mathbf{Q} \right) \right) \cdot \boldsymbol{\sigma} \quad (\text{R4})$$

where v_F^0 is the non-interacting Fermi velocity of graphene, and $U_d(\mathbf{Q})$ is the Fourier transformed superlattice potential at reciprocal vector \mathbf{Q} , which is expressed in Eq. (2) of main text (also see Eq. (R1)). From Eq. (R4), we note that the renormalized non-interacting effective Hamiltonian as well as the renormalized Fermi velocity have qualitatively the same dependence on the superlattice constant L_s for all the lattice geometries. Therefore, given the same L_s , the Fourier transformed superlattice potential $U_d(\mathbf{Q})$ of different lattice geometry has the same order of magnitude, and would only differ by a geometry-related factor of order of unity.

The above argument can be more explicitly illustrated as follows. Given a smooth superlattice potential in real space, its Fourier component $U_d(\mathbf{Q})$ decays exponentially with $|\mathbf{Q}|$, thus we only need to keep the terms with $\mathbf{Q} \in \{\mathbf{Q}_0\}$, where $\{\mathbf{Q}_0\}$ are collections of the primitive reciprocal vectors of the superlattice. For example, for rectangular lattice $\{\mathbf{Q}_0\} = \pm\mathbf{Q}_x, \pm\mathbf{Q}_y$ with $\mathbf{Q}_x = (2\pi/L_x, 0)$ and $\mathbf{Q}_y = (0, 2\pi/L_y)$; for triangular lattice $\{\mathbf{Q}_0\} = \{\pm\mathbf{Q}_1, \pm\mathbf{Q}_2, \pm(\mathbf{Q}_1 + \mathbf{Q}_2)\}$ with $\mathbf{Q}_1 = 2\pi/L_s(1/\sqrt{3}, -1)$ and $\mathbf{Q}_2 = 2\pi/L_s(1/\sqrt{3}, 1)$. Accordingly, the effective non-interacting Hamiltonian for rectangular lattice is:

$$H_{\text{rect}} = \hbar v_F^0 \left[\left(1 - 4 \frac{|U_d(\mathbf{Q}_y)|^2}{(\hbar v_F^0)^2 |\mathbf{Q}_y|^2} \right) k_x \sigma_x + \left(1 - 4 \frac{|U_d(\mathbf{Q}_x)|^2}{(\hbar v_F^0)^2 |\mathbf{Q}_x|^2} \right) k_y \sigma_y \right] \quad (\text{R5})$$

where the Klein tunneling effect manifests as the Fermi velocity in, e.g., the x-direction is reduced because of the potential scattering in the y-direction. For triangular lattice, the effective non-interacting Hamiltonian becomes:

$$H_{\text{tri}} = \hbar v_F \left(1 - 6 \frac{|U_d(\mathbf{Q}_1)|^2}{(\hbar v_F^0)^2 |\mathbf{Q}_1|^2} \right) \mathbf{k} \cdot \boldsymbol{\sigma} \quad (\text{R6})$$

where we clearly see the resemblance with H_{rect} , especially when the anisotropy parameter $r = L_y/L_x$ is on the order of 1.

We have also directly calculated the effective fine-structure constants $\alpha = e^2/(4\pi\epsilon_0\epsilon_r\hbar v_F)$ for both triangular and square lattices, where the renormalized Fermi velocity v_F is dependent on both superlattice constant L_s and relative dielectric constant ϵ_r . As shown in Fig. R4, the colormap of α for both triangular lattice (Fig. R4(a)) and square lattice (Fig. R4(b)) are similar to that of rectangular lattice given in the main text (Fig. 1(d)).

Fig. R4 Effective fine structure constant α for (a) triangular and (b) square superlattice potential, in which the dashed line marks the critical value ~ 0.92 for gap opening in graphene by interactions according to Ref. [32] of the main text.

In response to the reviewer's question, we have added a new paragraph discussing the universality of our results starting from line 89 in the resubmitted manuscript, where Eq. (R4) has been included into the main text as Eq. (3). We have also included Fig. R4 as Fig. S2 and the corresponding discussions at the end of the second last paragraph in Sec. S1 of Supplementary Material:

"It turns out that the colormaps of α remains qualitatively the same for different lattice geometries, comparing Fig. S1(a) for rectangular lattice with Fig. S2 for triangular and square lattice. This also justifies why we can consider only two particular cases, rectangular and triangular lattice, in our work without loss of generality."

Q&C 2. In DFT calculations, the atomic lattices of the insulating substrate must be commensurate with graphene carbon lattices. However, the requirement of this commensurability is lifted in the continuum model Eq.(1). If I understand correctly, the commensurability is not required and not important if the superlattice wavelength is much larger than carbon atomic lattice constant, and this condition is always satisfied throughout the paper.

Reply: We thank this comment from the reviewer. The reviewer understands deeply our theoretical modelling.

As mentioned by the reviewer, we consider a continuum model whose validity relies on the large superlattice wavelength. In practice, the lattice constant of superlattice is determined by transferred charge density in substrate. According to experiments, this density is at most $\sim 10^{13} \text{ cm}^{-2}$, which corresponds to a superlattice constant $L_s \sim 40\text{-}50 \text{ \AA}$, much larger than the lattice constant of graphene, $a = 2.46 \text{ \AA}$. So, we could legitimately use a continuum model throughout the paper, neglecting commensurability of the atomic lattices.

In response to the reviewer's comment, we have added the following sentence starting from line 73 of the resubmitted manuscript:

"Such a continuum-model description is adopted throughout the paper given that $L_s \gg a$ is always fulfilled for low carrier density $< \sim 10^{13} \text{ cm}^{-2}$, with $L_s \sim 1/\sqrt{n}$ for the EC state. "

Q&C 3. There's a tiny energy shift in red dashed lines in fig.2(c) compared to that in fig.2(b). The single-particle band structure should not depend on the filling factor.

Reply: We thank this comment from the reviewer.

The small energy shift between the non-interacting bands of Fig. 2(c) and Fig. 2(b) comes from our artificial definition that the chemical potential is fixed to zero energy. At charge neutral point, the Dirac point is pinned to zero energy; when graphene is hole-doped, the Dirac point is above zero energy.

We excuse ourselves for the lack of an explicit definition of zero energy in the band structure plots. To avoid such confusion and in response to the reviewer's comment, we have added the zero-energy convention in the caption of Fig. 2:

"Zero energies in (b) and (c) are defined as the chemical potentials for $\nu = 0$ and $\nu = -0.003$, respectively. "

Q&C 4. There's missing text In line 121, 'which will be discussed in detail in Sec. .'

Reply: We thank the reviewer's patience and efforts to help improve the readability of our manuscript. We have added the missing section number.

Reviewer #3 (Remarks to the Author):

The authors studied the interaction between a single-layer graphene and an insulator substrate. The electron crystal formed in the insulating layer provided a superlattice potential for graphene, resulting in the reduction of Fermi velocity. Furthermore, the authors considered the electron-electron interactions in graphene which led to gap opening at the Fermi level, and the non-trivial topology along with. While the numerical calculations and the theoretical analysis may provide better insight into the graphene-related research, the relation between the computation results and the conclusions is not decisive, and the significance of the research may not be high enough for publication in Nature Communication. Particularly, there are several issues that needs clearer illustration:

Reply: We thank the reviewer for the very careful reading of our manuscript and for the concise summary of our work. We feel sorry that the current version of the manuscript makes the reviewer find our conclusions lack of clear evidence to support them. To make our work more persuasive, we have added new results of calculations to provide stronger argument for the conclusions. We also have made significant changes both to the main text and the Supplementary Material, which are listed at the end of the reply, to render the paper more readable. Concerning the questions and comments posed by the reviewer, we will answer them point-to-point. We hope that the added results and discussions in the new version of the manuscript, as well as the clarification of the originality of our work can convince the reviewer to recommend the publication of our work.

Q&C 0. Part of the results have been reported by the authors' previous paper in Nat. Nano.

Reply: We first thank the reviewer for noticing our previous work published in Nat. Nano.. This paper is an experimental work done by our collaborators on an unexpected low-field, high-temperature quantum Hall effect discovered in graphene-CrOCl heterostructure, in which we suggest an explanation from the theoretical aspect.

In the experimental paper, the theoretical results (summarized in Fig.4 of the experimental paper) only give a flavor of the general idea. Whenever the idea of Wigner-crystal formation in the substrate and the interaction-driven Fermi-velocity enhancement in graphene were mentioned in the experimental paper, our independent theory preprint (which is a legacy version of the present manuscript) were always referred. In fact, Nat. Nano. accepted the experimental paper at the end of September 2022 after adding our theoretical support, knowing that our theoretical preprint had been uploaded to arXiv in June 2022 (*arXiv:2206.05659*), three months prior to the acceptance in Nat. Nano. In other words, the original idea that the charges transferred to the substrate form a Wigner crystal, which in turns boosts the interaction-driven gap opening and Fermi-velocity enhancement in graphene, was first proposed in our theory preprint *arXiv:2206.05659*. All discussions about this idea in the experimental Nat. Nano. paper have properly cited our theory preprint, which is a legacy version of the present manuscript.

Moreover, besides the original idea of charge-transfer induced Wigner crystal in the substrate and subsequent correlated states in graphene, in the present paper, we have presented a number of additional original results:

(a) For the first time we derived the general Hamiltonian describing such coupled graphene-insulator heterostructure systems (Eqs. (5a-e) of the main text).

(b) We constructed a general workflow to solve the ground state of such interacting coupled heterostructure system (see Methods), and discussed in depth how the gap is opened by interactions and how the Fermi velocity is enhanced in graphene (see the section "Coulomb Interactions in Graphene").

(c) We proposed topologically nontrivial subbands tuned by anisotropy of the superlattice (see section "Topological Properties").

(d) We discussed the stabilizing effect to the Wigner crystal state due to interlayer Coulomb interactions (see section "Cooperative Coupling between Graphene and Substrate").

(e) We performed high-throughput search of desirable insulating substrates to realize these effects (Table I), and some of our proposals, e.g., the CrI₃-graphene heterostructure, has already been experimentally realized.

None of the above results/ideas were mentioned in the experimental Nat. Nano. paper. Therefore, we believe that our theoretical work warrants another independent publication in Nature Communications.

Q&C1. The authors assumed from the very beginning that the insulating layer with rotationally symmetric 2DEG dispersion will form electron crystal phase with rectangular lattice. They did Hartree-Fock calculations to show that the total energy is lower than electron liquid phase. However, electron crystal is a strongly correlated system, and Hartree-Fock calculations are not enough to describe 1. how the Wigner crystal melts, and 2. how graphene layer stabilizes the crystalline phase. The single-particle plane wave basis is not an appropriate starting point. Furthermore, rather than assuming the rectangular shape of the lattice without reasoning, the authors should compare the ground state energy among different real space configurations, triangular, hexagonal, rectangular, etc. Still, the estimation of condensation energy requires careful treatment of many-body correlations, especially when the authors are trying to give quantitative estimate ('extrapolation' in FIG. 4) of the critical density for phase transition to the electron liquid.

Reply: We greatly appreciate the reviewer's critique. First, we completely agree with the reviewer that one needs to carefully check the lattice geometries of the Wigner crystal state. Previous studies (both quantum Monte Carlo and Hartree-Fock) indicate that at given Wigner-Seitz radius r_s , the ground-state of Wigner crystal forms a triangular lattice. We thus perform additional calculations using triangular lattice geometry, and find that indeed they have lower energy than the rectangular lattice. Previously we considered a rectangular superlattice, because the atomic lattice of CrOCl is rectangular, and it was assumed that the Wigner crystal is more or less coupled with the atomic lattice of CrOCl, thus favoring a

rectangular electronic superlattice. This constraint is released in our new calculations, and the data in Fig. 4 of the main text has been replaced by the ones for triangular superlattice.

Second, we also agree with the reviewer that mean-field theory cannot properly describe the quantum melting of Wigner crystal and cannot accurately estimate the condensation energy due to the negligence of correlation energy. However, the stabilizing mechanism of Wigner crystal, which is the key message we would like to convey, does not require an accurate estimate of the correlation energy nor a precise description of the quantum melting process. What is really needed is just the ground charge density in the Wigner crystal state, which can be properly treated by unrestricted Hartree-Fock theory as has been adopted in literatures (e.g., see *Phys. Rev. B* 68, 045107 (2003) and *Phys. Rev. B* 84, 115115 (2011)). We explain this in detail in the following.

First, we have made an ansatz to the ground-state wavefunction of the coupled bilayer system, i.e., the “separable wavefunction ansatz” (see Eq. (6) of the main text):

$$|\Psi\rangle = |\Psi\rangle_c \otimes |\Psi\rangle_d, \quad (R7)$$

which says that the ground state of the coupled bilayer system $|\Psi\rangle$ can be written as the product of the wavefunction in the graphene layer $|\Psi\rangle_c$ and that of the substrate’s layer $|\Psi\rangle_d$. Under this ansatz, the ground-state expectation value of $\langle \hat{c}^\dagger \hat{d} \rangle$ and $\langle \hat{d}^\dagger \hat{c} \rangle$ vanish (\hat{c}/\hat{c}^\dagger and \hat{d}/\hat{d}^\dagger are the annihilation/creation operators for electrons in graphene and in the substrate, respectively). Since a nonzero $\langle \hat{c}^\dagger \hat{d} \rangle$ requires Coulomb scattering from graphene’s valley to the substrate’s valley (in reciprocal space), but electrons in the substrate and graphene are typically located at different atomic valleys in reciprocal space, such that the intervalley Coulomb scattering amplitude is several orders of magnitude weaker than the intravalley one (see discussions below Eq. (6) in the main text). For example, if the conduction band minimum of the charge doped substrate is at Γ point, while Dirac point is at \mathbf{K} , then the intervalley Coulomb scattering amplitude $\sim e^2 \exp(-|\mathbf{K}|d) / (2\epsilon_0\epsilon_r|\mathbf{K}|)$, which is several orders of magnitude weaker than the intravalley Coulomb scattering amplitude $\sim e^2 \exp(-|\mathbf{Q}|d) / (2\epsilon_0\epsilon_r|\mathbf{Q}|)$, where d is the interlayer distance ($d \approx 7 \text{ \AA}$ for CrOCl-graphene according to DFT calculations), $|\mathbf{Q}| \sim 2\pi/L_s$ (L_s is superlattice’s constant), while $|\mathbf{K}| = 4\pi/3a$ (a is graphene’s lattice constant). It turns out that for the superlattice constants of interest, the intervalley Coulomb scattering amplitude $\sim 10^{-8}$ - 10^{-7} eV, which is completely negligible. Thus, we see that Eq. (R7) (Eq. (6) in the main text) is justified as long as the valleys of the two layers are well separated in reciprocal space such that the intervalley Coulomb scattering is negligible.

With the separable wavefunction hypothesis, the expectation value of the interlayer Coulomb interaction term can always be expressed as:

$$\langle H_{\text{gr-sub}} \rangle = \sum_{\mathbf{q}} \frac{e^2 e^{-|\mathbf{q}|d}}{2\epsilon_0\epsilon_r\Omega_d|\mathbf{q}|} \rho_d(\mathbf{q}) \rho_c(-\mathbf{q}) \quad (R8)$$

where

$$\rho_c(-\mathbf{q}) = \frac{1}{N_c} \sum_{\{\mathbf{k}, \mu, \sigma, \alpha\}} \langle \hat{c}^\dagger_{\sigma\mu\alpha, \mathbf{k}-\mathbf{q}} \hat{c}_{\sigma\mu\alpha, \mathbf{k}} \rangle \quad (R9a)$$

$$\rho_d(\mathbf{q}) = \frac{1}{N_c} \sum_{\{\mathbf{k}, \sigma\}} \langle \hat{d}^\dagger_{\sigma, \mathbf{k}+\mathbf{q}} \hat{d}_{\sigma, \mathbf{k}} \rangle \quad (R9b)$$

are just the Fourier transformed charge densities in the graphene layer and in the substrate, respectively. Here μ, σ, α denote valley, spin, sublattice degrees of graphene, \mathbf{k} and \mathbf{q} are the wavevectors, and N_c is the number primitive cells in the entire system. Again, Eqs. (R8) and (R9) are correct as long as the separable wavefunction ansatz is valid, which is perfectly justified due to the negligible intervalley Coulomb scatterings as discussed above. We impose no more constraints on the form of the wavefunctions $|\Psi\rangle_c$ and $|\Psi\rangle_d$.

Now, if electrons in both graphene and the substrate stay in Fermi-liquid states with homogeneous charge densities, then $\rho_c(-\mathbf{q}) = \rho_c(0)\delta_{\mathbf{q},0}$, and $\rho_d(\mathbf{q}) = \rho_d(0)\delta_{\mathbf{q},0}$; if the electrons in the substrate stay in Wigner crystal state with superlattice constant L_s and reciprocal lattice vector \mathbf{Q} , then the real space charge density has to obey superlattice translational symmetry, such that $\rho_d(\mathbf{q}) = \sum_{\mathbf{Q}} \rho_d(\mathbf{Q})\delta_{\mathbf{q},\mathbf{Q}}$. Similarly, since the graphene's layer is Coulomb coupled to the Wigner crystal, graphene's charge density obeys the same kind of superlattice modulation which also satisfies, $\rho_c(-\mathbf{q}) = \sum_{\mathbf{Q}} \rho_c(-\mathbf{Q})\delta_{\mathbf{q},\mathbf{Q}}$. As a result of the superlattice translational symmetry, in the Wigner crystal state (coupled with graphene), the interlayer Coulomb energy is changed with respect to the Fermi-liquid state by an amount

$$\delta\langle H_{\text{gr-sub}} \rangle = \sum_{\mathbf{Q} \neq \mathbf{0}} \frac{e^2 e^{-|\mathbf{Q}|d}}{2\epsilon_0 \epsilon_r \Omega_d |\mathbf{Q}|} \rho_d(\mathbf{Q}) \rho_c(-\mathbf{Q}). \quad (\text{R10})$$

Then, as we discussed in both the main text and in Sec. S6 of Supplementary Material, the charge centers of the two layers can be shifted with respect to each other by a vector \mathbf{t}_m , and the interlayer Coulomb energy can be minimized by choosing an optimal \mathbf{t}_m . As expected, the optimal \mathbf{t}_m is determined such that the charge centers of the two layers are pinned to each other in an anti-phase interlocked pattern (see Fig. 4(a)-(b) of the main text). This is precisely how the Wigner crystal state is stabilized: at some Wigner-Seitz radius $r_s < r_s^*$ ($r_s^* \sim 31$ is the critical value for Wigner crystal transition for free 2D electron gas according to quantum Monte Carlo calculations), the total energy of the Fermi-liquid state (E_{FL}) is supposed to be lower than that of a Wigner crystal (E_{WC}), i.e. $\Delta = E_{\text{WC}} - E_{\text{FL}} > 0$. Then, the interlayer Coulomb energy would exactly compensate this energy, such that $\Delta + \delta\langle H_{\text{gr-sub}} \rangle < 0$, which thus postpones the Wigner crystal transition to a smaller critical r_s and a larger critical carrier density.

The above argument only requires two prerequisites: **(a)** the separable wavefunction ansatz, Eq. (R7), which is valid given that the intervalley Coulomb scattering is negligibly weak; and **(b)** the superlattice translational symmetry of the Wigner crystal such that the interlayer Coulomb energy can be written in the form Eq. (R10). No further assumption is made on the ground-state wavefunctions. Therefore, the idea of Wigner crystal stabilized by interlayer Coulomb interaction remains valid even if the ground-state wavefunction is more complicated than Slater determinant, i.e., beyond Hartree-Fock treatment.

What we did in this work is to further assume that $|\Psi\rangle_c$ and $|\Psi\rangle_d$ are Slater determinants, i.e., the Hartree-Fock approximation. We agree with the reviewer that this is a drastic approximation which does not properly take into account the correlation energy, thus cannot estimate the Wigner crystal condensation energy accurately. However, as discussed

above, the stabilizing mechanism of Wigner crystal (by interlayer Coulomb interaction) only requires a proper estimate of the ground-state charge density of the Wigner crystal, which can be decently treated by Hartree-Fock method, e.g., see *Phys. Rev. B* 68, 045107 (2003) and *Phys. Rev. B* 84, 115115 (2011).

Nevertheless, in order to fully resolve the reviewer's concern, we have adopted the fitting model for spin polarized Wigner crystal energy of 2D electron gas introduced in *Phys. Rev. Lett.* 102.126402 (2009):

$$E_{WC} = \frac{c_1}{r_s} + \frac{c_{3/2}}{r_s^{3/2}} + \frac{c_2}{r_s^2} + \frac{c_{5/2}}{r_s^{5/2}} + \frac{c_3}{r_s^3} \quad (R11)$$

where $c_1 = -1.106103$, $c_{3/2} = 0.814$, $c_2 = 0.113743$, $c_{5/2} = -1.184994$, and $c_3 = 3.097610$ and the total energy is given in Hartree atomic units. These parameters are determined by fitting to quantum Monte Carlo data. The total energy for the Fermi-liquid state is given by the following model, which is also determined by fitting to quantum Monte Carlo data, given in *Aust. J. Phys.*, 1996, 49, 161-82:

$$E_{FL} = E_{FL}^{hf} + E_{FL}^c \quad (R12a)$$

$$E_{FL}^{hf} = \frac{1}{2r_s^2} - \frac{4\sqrt{2}}{3\pi r_s} \quad (R12b)$$

$$E_{FL}^c = a_0 \left\{ 1 + A x^2 \left[B \ln \frac{x+a_1}{x} + C \ln \frac{\sqrt{x^2+2a_2x+a_3}}{x} + D \left(\arctan \frac{x+a_2}{\sqrt{a_3-a_2^2}} - \frac{\pi}{2} \right) \right] \right\} \quad (R12c)$$

where $x = \sqrt{r_s}$ and:

$$A = \frac{2(a_1+2a_2)}{2a_1a_2-a_3-a_1^2}, \quad B = \frac{1}{a_1} - \frac{1}{a_1+2a_2}, \quad C = \frac{a_1-2a_2}{a_3} + \frac{1}{a_1+2a_2},$$

$$D = \frac{F-a_2C}{\sqrt{a_3-a_2^2}}, \quad \text{and } F = 1 + (2a_2 - a_1) \left(\frac{1}{a_1+2a_2} - \frac{2a_2}{a_3} \right) \quad (R12d)$$

with $a_0 = -0.1925$, $a_1 = 7.3218$, $a_2 = 0.16008$, and $a_3 = 3.1698$. The energies are given in Hartree atomic units.

From the above equations, we can determine the condensation energy of the Wigner crystal state (denoted by $\Delta = E_{WC} - E_{FL}$) to the accuracy of quantum Monte Carlo calculations. Then we include the minimized interlayer Coulomb interaction energy $\delta\langle H_{gr-sub} \rangle$ (by choosing an optimal relative vector \mathbf{t}_m) using Eq. (R10), where the charge density of the two layers are calculated using unrestricted Hartree-Fock approximations. In particular, a triangular superlattice has been adopted as it is the actual ground state for the Wigner crystal. We further calculate the ground-state energies of graphene layer for the gapped (denoted by $E_{gap,gr}$) and gapless state (denoted by $E_{gapless,gr}$) with unrestricted Hartree-Fock treatment. Then, the total energy gain for the Wigner-crystal state in the substrate coupled with gapped Dirac state in graphene is:

$$\Delta_{tot} = \Delta + \delta\langle H_{gr-sub} \rangle + E_{gap,gr} - E_{gapless,gr} \quad (R13)$$

In Fig. R5(c), we plot both Δ_{tot} per electron (orange line) and Δ (with the Wigner-crystal condensation energy Δ determined by quantum Monte Carlo calculations, green line) as

a function of carrier density. Clearly, the Wigner crystal state becomes significantly stabilized by virtue of the interlayer coupling. The critical carrier density is postponed from $4.5 \times 10^{12} \text{ cm}^{-2}$ (corresponding to $r_s = 32.9$) to $16 \times 10^{12} \text{ cm}^{-2}$ (corresponding to $r_s = 17.3$), where an effective mass $m^* = 1.3m_0$ (m_0 is the bare electron mass), a background dielectric constant $\epsilon_r = 4$, and twofold valley degeneracy have been adopted (to mimic the conduction band minimum of CrOCl). The quantum critical regime (with condensation energy ~ 0) is beyond the scope of the present study.

Fig. R5 Charge density modulations after minimizing interlayer Coulomb interactions for (a) gapped Dirac state in graphene, and (b) electronic-crystal state in the substrate, which forms triangular superlattice. (c) Condensation energy (per electron) of the electronic crystal state E_{cond} vs. the carrier density n in the substrate. The green line represents the condensation energy of the decoupled system using the quantum Monte Carlo data, and the orange lines shows that of the coupled bilayer system, which is significantly stabilized by interlayer Coulomb potential. The critical r_s for Wigner-crystal transitions are also indicated in the coupled and decoupled cases, respectively.

The reviewer is also concerned that plane-wave basis is not a suitable basis to describe the Wigner crystal state. We are sorry that we did not explain this very clearly in the previous version of the manuscript. When calculating the ground-state charge density of the Wigner crystal using Hartree-Fock approximation, a 9×9 mesh of reciprocal lattice points have been adopted, with the Brillouin zone sampled by a 18×18 k mesh. The former corresponds to the real-space mesh within a primitive cell, while the latter corresponds to the system size. In Fig. R6 we show the real-space charge density of the Wigner crystal calculated using different real-space mesh (from left to right, 9×9 , 11×11 , 13×13 , 15×15). We see that the charge density is quite localized in real space, and the calculation is converged already with a 9×9 real-space mesh. We have also reproduced our calculations at several values of carrier densities using a 13×13 real-space mesh, and obtain fully converged results.

Fig. R6. Ground-state charge density for the spin polarized Wigner crystal state (with triangular lattice) of free 2D electron gas with carrier density $6.6 \times 10^{12} \text{ cm}^{-2}$, effective mass $m^* = 1.3 m_0$ (m_0), background dielectric constant $\epsilon_r = 4$, and a valley degeneracy of 2, corresponding to the parameters of CrOCl conduction band minimum. From left to right: charge densities calculated with 9×9 , 11×11 , 13×13 , and 15×15 reciprocal-space cutoff (real-space mesh).

Fig. R7 Graphene-CrOCl heterostructure's non-interacting bands, in which we project the orbital contribution on the two sides of the heterostructure: graphene's C atoms (left) and CrOCl's Cr atoms (right). Here, we slightly alter graphene's lattice to fit in a commensurate superlattice of CrOCl. More details for the commensurate supercell construction are given in the need added subsection of Sec. S7 of Supplementary Material (page 24). The lattice distortion amounts to adding strain to graphene and thus moves the position of Dirac cone away from high-symmetry lines. Therefore, we choose the line (from Γ to M' , which is not a high-symmetry point but near M) that passes the Dirac point in the Brillouin zone to plot the band structure.

In response to the reviewer's questions and comments, we have replaced the condensation energy vs. carrier density plot in Fig. 4 by Fig. R5, such that the condensation energy of Wigner crystal now is determined by the data from quantum Monte Carlo calculations. We also have rewritten the caption accordingly. We have added a few paragraphs (starting from line 380 to line 390) and Eqs. (11)-(13) to describe the model of condensation energy of Wigner crystal, which is determined by fitting to quantum Monte Carlo data.

We have also added a few sentences starting from line 276 of the main text:

"The energy difference between the spin polarized EC state and Fermi-liquid (FL) state (condensation energy) as a function of the carrier density n is given by quantum Monte Carlo calculations in [55,56], as shown by the green line in Fig. 4(c). The condensation energy reaches zero when $n \approx 4.5 \times 10^{12} \text{ cm}^{-2}$ suggesting the transition from the EC to the FL state."

We have also added the following sentences starting from line 286 of the main text:
"Specifically, with the separable wavefunction ansatz (Eq. (6)), the ground-state charge densities for the graphene layer and the EC layer are obtained from Hartree-Fock calculations, which are further used to estimate the interlayer Coulomb energy."

Q&C 2. If there is charge transfer between the layers, why can one still treat the electrons in two layers separately, and consider the substrate layer simply as Dirac-delta potentials?

Reply: We thank the reviewer for asking this important point on our theoretical modelling. We would like to answer it in two folds.

First, the strength of interlayer hopping between graphene and substrate, which relies on short-ranged atomic orbital overlapping and exchange effects, decays exponentially with the interlayer distance so that it is weak and thus negligible with large interlayer distance (~ 7 Å for CrOCl-graphene heterostructure according to DFT calculations). This is further confirmed by our new DFT calculations of the orbital projected band structures of CrOCl-graphene heterostructure, as shown in Fig. R7. The Dirac cone in the band structure stems almost entirely from C atoms of graphene. The conduction band from Cr atoms in CrOCl is nearly intact while interfacing with graphene. So, we can neglect any short-ranged interlayer coupling and focus on long-ranged interactions via interlayer Coulomb potential.

The charge transfer is driven by the quantum tunneling effects, i.e., the charges in the two layers would redistribute to a global equilibrium ground state. During such a process, electrons would tunnel from one layer to the other, with a characteristic time scale $\sim \hbar/w$, where w is the interlayer hopping amplitude, which is negligibly small as discussed above and shown in Fig. R7, but still not rigorously zero. The charge transfer process is established after a time interval $\sim \hbar/w$, then the equilibrium ground state is reached. Assuming an extremely small interlayer hopping amplitude $w=0.1$ meV (which is several orders of magnitudes weaker than the interlayer Coulomb energy), the characteristic time scale for the tunneling process is on the order of picosecond, as has been experimentally measured for graphene-WS₂ heterostructure (*Nat. Commun.* 5, 5622 (2014)).

Nevertheless, according to Eq. (5e) in the manuscript (Eq. (4e) in the previous version of the manuscript), electrons from the two layers are indistinguishable so that the treatment of interlayer Coulomb interactions are still difficult. This is where the separable wavefunction hypothesis sets in. The latter, which also can be called "Hartree factorization" allows to treat electrons in two layers separately. Indeed, "Fock factorization" of the interlayer Coulomb interactions should be in principle also taken into account. It turns out that, as briefly explained in the paragraph below Eq. (6) in the manuscript (and explained in detail in the reply to Q&C 1 from Reviewer 3), the Fock-type term is governed by large momentum transfer due to the large mismatch between Dirac points sits at \mathbf{K} point in graphene's Brillouin zone and the relevant band edge of substrate in its proper Brillouin zone. For CrOCl, the band edge sits on the Γ -X line, as shown in Fig. S14(a). Since the

interlayer Coulomb potential decays exponentially with momentum transfer (see our reply to Q&C 1 from the Reviewer 3), we can neglect such effect in most of substrate materials. This is the reason why we could treat the two layers separately.

However, the reviewer is totally right about that such effect has to be considered when the mismatch between Dirac points in graphene and the band edge in substrate is tiny, for example in some of the transition metal dichalcogenides where the low-energy band edge also sits at K points in its Brillouin zone. This would enable the emergence of a new phase characterized by the Fock factorization $\langle c^\dagger d \rangle$ (or $\langle d^\dagger c \rangle$). We call such new state of matter *interlayer excitonic condensate* because it consists of Dirac holes (electrons) in graphene and quadratically dispersive electrons (holes) in substrate, as implied by the name. Although this phase is interesting, it is beyond the scope of this theoretical study and we leave it to our immediate next work.

The reviewer's second concern is about the interlayer Coulomb potential given by Eq. (2) in our modelling. Since the separate wavefunction hypothesis is valid as explained above, we can integrate the degrees of freedom of substrate into a superlattice potential. The form of potential in Eq. (2) amounts to considering the electronic density of Wigner crystal formed in the substrate as a sum of Dirac delta functions arranged in lattice. In practice, the distribution of electronic density is of course not peaked as Dirac delta but smooth in space. So, in our second modelling in the section "Cooperative Coupling between Graphene and Substrate", we relax this restriction and consider a more realistic charge distribution. It turns out that the interlayer Coulomb potential arising from realistic charge distribution Eq. (10) in the main text has the same functional form as Eq. (2). This shows that our first modelling already captures the essential physics. More details of the derivation of Eq. (10) is given in Sec. S6 of Supplementary Material.

In response to the reviewer's questions. We have added the following sentences starting from line 253 in the main text:

"This is further confirmed by directly calculating the orbital projected band structures of a commensurate supercell of CrOCl-graphene heterostructure based on density functional theory. It turns out that the Dirac cone in such heterostructure supercell stems almost 100% from carbon p_z orbitals of graphene (see Sec. S7 of Supplementary Material), which clearly indicates the absence of interlayer hybridization (hopping)."

We have included Fig. R7 (Fig. S15 in Supplementary Material) and the corresponding discussions into the second subsection of Sec. S7 of Supplementary Material.

We have also included the following sentences starting from line 272 of the main text, which briefly discuss the possibility of interlayer excitonic insulator in valley-matched graphene-insulator heterostructures:

"Nevertheless, the interlayer excitonic insulator state consisted of Dirac electrons (holes) and quadratically dispersive holes (electrons) is possible in valley-matched graphene-insulator heterostructures, such as those consisted of graphene and transition metal dichalcogenides with the band extrema at K points. We leave this for future study."

Q&C 3. Line 152. What is special about these 'hot spots' that contribute to large Berry curvature? Is there a way to identify their nature and how it changes with L_s and r ? The origin of new topological properties is not demonstrated clearly.

Reply: We thank the referee for asking this question regarding the topological aspects of the system.

The band crossings via tuning superlattice constant L_s and anisotropy r are accidental in the sense that no symmetry protects it. Therefore, these crossings are gapped once L_s and r deviate from the fine-tuned values, and they can appear in principle anywhere in the Brillouin zone. These band crossings with concentrated Berry curvature originate from high-symmetry points, which plays the role of the source of Berry curvature. Every time two bands cross, their Berry curvature are exchanged between them and the associated Chern number is altered, i.e., the topological transition occurs by the band inversion mechanism. If we monitor where these band crossings originate and how they are gapped out by tuning L_s and r , we can identify the nature of the topological transitions.

To illustrate this point, we show this process in six videos as Supplementary Movie S1 to S6 (noted as SMS1 to SMS6): SMS1 and SMS2 for $L_s = 50 \text{ \AA}$, SMS3 and SMS4 for 200 \AA , SMS5 and SMS6 for 600 \AA , respectively. Note that these six videos are those that were failed to be uploaded in last submission. Given L_s , videos with odd number are collections of band structures varying anisotropy parameter r . Those with even number show the distribution of Berry curvature as a function of r . For example, for $L_s = 50 \text{ \AA}$, the video SMS1 shows the evolution of the band structure while increasing the anisotropy parameter r from 1 to 10. The other one SMS2 shows in parallel the evolution of Berry curvature distribution in the Brillouin zone during this process. Here, we provide two animated images in Fig. R7 showing the emergence of a crossing point from Dirac point for $L_s = 50 \text{ \AA}$ and r ranging from 1 to 5. As shown in the left panel of the animated Fig. R8, as $r = L_y/L_x$ starts to increase, the Fermi velocity in the k_x -direction is gradually reduced, which can be explained by anisotropic Fermi velocity renormalization induced by the rectangular superlattice potential scattering, as expressed in Eq. (R5) in the reply to Q&C1 of Reviewer 2. At $r=2.7$, the Fermi velocity along the k_x -direction becomes vanishingly small. Then, further increasing r to 2.8 germinates a band crossing (indicated by red circle) originated from the Dirac point (Γ_s) then gradually moving away from Γ_s towards X_s point. At exactly the same moment that bands start to cross, a hot spot of Berry curvature appears (also indicated by red circle in the right panel of Fig. R8) and follows the trajectory of the crossing point with increasing r . Note that Fig. R8 is also presented in Fig. S4 of Supplementary Material.

Nevertheless, the Dirac point remains gapless during this process since its protecting symmetry $C_{2z}T$ has never been broken in our model Hamiltonian. In practice, in the non-interacting framework, we have to introduce a tiny small sublattice gap (0.001 meV, way smaller than any other energy scale in the system) to break the above symmetry so as to soundly define valley Chern number. The introduction of such a gap is a posteriori justified

by the Hartree-Fock calculation including e-e interaction effects. The gap opened by interactions for graphene at CNP is a sublattice gap (see Table R1) so that $C_{2z}T$ symmetry is spontaneously broken. Alternatively speaking, e-e interactions would add automatically a sublattice gap to graphene at CNP and the resulting ground state is topologically equivalent to the non-interacting state with a sublattice gap.

Fig. R8 Non-interacting band structures (left) and Berry curvature distribution (right) of graphene on a superlattice with $L_s = 50 \text{ \AA}$ and the anisotropy parameter r varying from 1 to 5. □ “hot spot” starts to appear at $r=2.8$, which is annotated by red circle. Note that this is an **animated figure**. In case of problem, we also include the same animated figure in Supplementary Material as Fig. S4.

In response to the reviewer’s comments, we have added the above discussion on the topological properties with respect to different lattice parameters in the main text. Starting from line 192, we have added a few sentences in the main text:

“...to create an accidental crossing point”

and in line 194:

“On the one hand, the band crossing moving away from Γ_s is of accidental nature, which is generally avoided unless the lattice parameters are at some fine-tuned values. On the other hand, the Dirac point at Γ_s remains stable as protected by $C_{2z}T$ symmetry.”

Besides, we have appended the animated Fig. S4 and six videos as supplemental data.

Q&C 4. In supplement S1, ‘The sum over Q in Eq. (S23) stops at the limit $|n_x| + |n_y| \leq 2$. What is the justification of not considering larger number of folded zones?’

Reply: We thank the referee for asking this technical question. The interlayer Coulomb potential has an exponential decaying form as a function of momentum transfer \mathbf{Q} , as shown in Eq. (2). So, the contribution of large \mathbf{Q} in Eq. (S23) of Supplementary Material is negligible. It turns out that the cut-off $|n_x| + |n_y| \leq 2$ manage to capture this effect. It keeps the following calculations numerically economic without losing key information.

Furthermore, we compare the band structures with the cut-off with those summing over

all the \mathbf{Q} vectors, for $L_s = 50 \text{ \AA}$ and $r = 1.2$, as shown in Fig. R9. We see that the two groups of bands exactly coincide with each other in most of places while only differ a little at high energy in the valence bands. Since we are interested in the low-energy physics of graphene-CrOCl heterostructure, the cut-off used in Eq. (S23) of Supplementary Material should affect little our calculations.

Fig. R9 Band structures of graphene on a superlattice potential given by Eq. (2) for $L_s = 50 \text{ \AA}$ and $r=1.2$. The bands using the cut-off $|n_x| + |n_y| \leq 2$ on \mathbf{Q} vectors are plotted using blue dashed lines and those considering a potential with all the \mathbf{Q} vectors are plotted using red dashed lines.

Q&C 5a. The authors show in S9 that with electrical transport measurement of the graphene-CrOCl heterostructure, gap opening at CNP can be observed from the longitudinal resistivity peak. They determined the gap size by fitting temperature-dependence of resistivity with $\exp(-\Delta/2kBT)$. The link between experiment and theory is poor. Firstly, the authors should show that the sample quality is good enough, so that the resistivity peak is indeed from a gapped phase. At least images of the experimental configuration and the measurement setup should be provided.

Reply: We thank the reviewer for asking this question on the experimental data and their interpretation.

As requested by the reviewer, we include below detailed experimental data on the quality of sample, the measurement setup and the device configuration.

First, a cartoon illustration of the tested device is showed in Fig. R9a, which includes h-BN/Graphene/CrOCl van der Waals heterostructure equipped with a top gate and a bottom gate. Fig. R9b and Fig. R9c show the 5X magnification and 100X magnification optical pictures of the device, respectively. There are no visible bubbles or wrinkles on the heterostructure. To ensure the cleanliness of the sample, we will use an AFM (atomic force microscope) tip to clean the Au bottom gate in a Contact mode before landing heterostructures onto these local metallic gates. A significant amount of PMMA can be

removed via such a process to make sure the homogeneity of the gate electrical fields (Fig. R9d and Fig. R9e). The device studied in this work is the same device S40 we investigated in our recent experimental paper published in *Nat. Nanotechnol.* **17**, 1272–1279 (2022). According to the high quality of quantum Hall plateaus (in the normal phase without charge transfer) seen at moderate magnetic field at a few or a few tens of Kelvin in these samples, we can claim that those samples, in term of cleanness, are of state-of-the-art quality.

In order to test the samples via transport, we use a wire bonder to bond the device onto the sample holder (Fig. R9a). Samples are then loaded into a 1.5 K fridge (Oxford Tesla-Tron system), with a base temperature of 1.5K and a maximum magnetic field of 12 T. We adopt standard 4-probe low frequency lock-in measurement method to perform electrical transport characterization of the devices, with the Stanford SR830 lock-in amplifier in series with a $10\text{ M}\Omega$ bias resistor, thus providing an AC signal at constant current of 100 nA (assume the sample resistance is way smaller than the bias resistor). Meanwhile, high precision voltage source (Keithley 2400) is used to regulate the top and bottom gate voltages. The longitudinal voltage V_{xx} is measured by the lock-in amplifier.

Fig. R10 The characterizations of the experimental devices. (a) The cartoon illustration of a typical h-BN/Graphene/CrOCl heterostructure device, with its optical pictures shown in (b) and (c). Scale bar in the image is $10\ \mu\text{m}$. Images of AFM scans of the bottom gate before (d) and after (e) AFM Contact mode cleaning are also illustrated, where the cleaned window can be highlighted by the PMMA residues accumulated during the AFM ‘sweeping’ cleaning.

Fig. R11 Measurement configurations. (a) The image of device bonded onto the holder. (b) Diagram of the 4-probe measurement method of Hall bar. The electrode labeled “ I_{ac} ” is applied 100nA AC signal by SR830 with a 10 M Ω bias resistor. The longitudinal voltage V_{xx} is recorded by SR830, as well. The top gate and bottom gate electrodes are supplied voltage by Keithley 2400. Scale bar in the image is 5 μ m.

In addition, according to our experimental observations, we actually have two phases (Fig. R12) of graphene when interfaced with few layered CrOCl, when measuring the sample resistance in the parameter space of V_{tg} and V_{bg} :

1. Phase-I is conventional graphene showing rather low Dirac peak at a few hundreds of Ohms;
2. Phase-II is the interfacial-coupling phase, where the band structure of graphene is re-configured with a mild gap opening at the charge neutral.

In fact, our theory is concerned about the Phase-II since Phase-I is the trivial conventional graphene while the Wigner crystal in the substrate is not activated since the charge transfer is not onset. Given that quantum Hall plateaus have been clearly seen under sub-Tesla magnetic field at tens of Kelvin, with the Shubnikov-de Haas quantum oscillations shown in Fig. R12a, the quality of sample is guaranteed. So, the measured resistivity peak has to do with gap opening in the Phase-II in CrOCl contacted graphene. For even more experimental details, we would invite our referee to kindly refer to the Supplementary Information of *Nat. Nanotechnol.* **17**, 1272–1279 (2022), and to its Supplementary Information available at:

https://static-content.springer.com/esm/art%3A10.1038%2Fs41565-022-01248-4/MediaObjects/41565_2022_1248_MOESM1_ESM.pdf

In response to the reviewer’s comments, we have added the following sentences in the main text at line 127:

“...using the same high-quality device reported in Ref. [19]. The details for the quality of the sample, the measurement set up, and the device configuration are presented in Supplementary Material (Sec. S8).”

Fig. R12. (a)-(b) Experimental and calculated phase diagram for the CrOCl contacted graphene in the parameter space of displacement field D and total effective gate-induced carrier density n . (c) Schematic cartoon illustrating the conventional gapless graphene (Phase-i), and gapped graphene due to interfacial coupling between graphene and the Wigner crystal in the surface state induced in CrOCl. \square and the formation of such Wigner crystal depends on the band alignment between the interfacial state and graphene, as illustrated in (d)-(j). Figure adopted from the Figure 4 in the main text of *Nat. Nanotechnol.* **17**, 1272–1279 (2022),

Fig. R13. Shubnikov-de Haas (SdH) analysis of typical dual-gated CrOCl/graphene/h-BN samples. (a)-(b) The Shubnikov-de Haas (SdH) oscillations from various temperatures at dopings of $n_{\text{tot}} \pm 2.5 \times 10^{12} \text{ cm}^{-2}$ and $D_{\text{eff}} = -0.4 \text{ V/nm}$ in Device-S40. (c) Lifshitz-Kosevich fit in Phase-i and Phase-ii in Device-S40, at 2.8 T and 1.7 T, respectively. (d) Cyclotron masses obtained in several samples. It is seen that the m^* in Phase-I in our system is in agreement with the “ordinary” monolayer graphene reported elsewhere. Figure adopted from the Supplementary Figure 36, in *Nat. Nanotechnol.* **17**, 1272–1279 (2022).

Q&C 5b. Secondly, there is no evidence of Wigner crystal formation, and the authors did not rule out any other possible factors that may lead to gap opening.

Reply: We try our best to rule out all the possibilities we can imagine. In the non-interacting framework, $C_{2z}T$ has to be broken to open a gap in graphene. The experiments have shown that the antiferromagnetism in CrOCl is irrelevant, because the resistivity peak and the $R_{xx} \sim \exp(\Delta_g/2k_B T)$ behavior remain intact above the Néel temperature ($\sim 14 \text{ K}$), where R_{xx} , Δ_g , and T denote resistivity, gap, and temperature, respectively. This is why

we resort to e-e interactions which may be able to spontaneously open a gap in graphene. However, in previous experimental works, no gap opening has been observed in graphene, free-standing or placed on an inert substrate as hBN. So, we infer that the gap opening cannot be solely due to the intralayer Coulomb e-e interactions in graphene: the substrate has to play a crucial role in graphene-CrOCl heterostructure, which is to reduce the non-interacting Fermi velocity in graphene to boost the e-e interaction effects as discussed throughout our manuscript.

Furthermore, the experimental results clearly show that charges (with carrier density $\sim 10^{12} \text{ cm}^{-2}$) tunnel from graphene to substrate, while the substrate still remains insulating for a continuum of density of transferred charges $< \sim 7 \times 10^{12} \text{ cm}^{-2}$. This proves that the transferred charges in the substrate form a new insulating state, which cannot be a band insulator. Instead, this should be an interaction-driven correlated insulator in the slightly charge doped substrate.

We then consider two types of correlated insulators: Mott insulator and spontaneous symmetry-breaking one. The first one is ruled out because the substrate itself without any doping may already be a Mott insulator, e.g., in the case of CrOCl, but the gap in graphene can only be observed when it is charge doped. Therefore, we believe that the correlated insulating state in substrate belongs to the second possibility. We focus on the simplest case where the spatial translational symmetry is broken and a long-wavelength charge order is formed. This charge order could be Wigner crystal, long-wavelength charge density wave, or something else. Actually, we model a long-wavelength charge order that onsets at small carrier density instead of limiting ourselves to Wigner crystal formation, which is the mechanism we proposed in the main text. While charge density wave formation is a case-by-case study, which relies crucially on the shape of Fermi surface and phonon softening etc., the formation of Wigner crystal is more generic only requiring a large Wigner-Seitz radius. As we are looking for universality of our theory, we discuss only the formation of Wigner crystal in the section related to material realizations. Luckily, high-output DFT calculations manage to find a series of relevant materials that could showcase the physics we discussed in the paper.

Nevertheless, we did not discuss in detail another possibility, namely interlayer excitonic insulator. An interlayer attraction between hole and electron in the two layers hinder the movement of carries leading to an excitonic-like insulating phase. However, the interlayer excitonic insulator is disfavored when the conduction band minimum of CrOCl is well separated from \mathbf{K} point of graphene, as already discussed in our reply to Q&C 1 from Reviewer 3. Therefore, after ruling out all these possibilities. We believe that the Wigner crystal boosted correlated insulator is the natural mechanism to explain the gap opening in graphene. Most saliently, our calculated gap agrees with the experimental measured gap very well, and this match (between theory and experiment) is even better after taking into account the valley degeneracy of the conduction band minimum of CrOCl (see Fig. R3).

Moreover, we are aware that recently an experimental group expertized in scanning

tunneling microscopy has already observed the signature of Wigner crystal state in one of such charge-transfer graphene-insulator heterostructures. Their results are still being polished, and are not published yet. We hope that this can fully address the reviewer's concern.

And some grammatical and arrangement issues:

1. Organization can be better. Too much repetition of long paragraphs between main texts and supplement.

Reply: We thank the reviewer for this kind suggestion. We have deleted a number of repetitive paragraphs and equations in Sec. S6 of Supplementary Material.

2. Line 121 'Sec..'

Reply: We thank the reviewer for pointing out the typo. This has been fixed in the resubmitted manuscript.

3. Line 194-202 are lengthy. Please refine the words.

Reply: Following the reviewer's suggestion, line 194-202 (line 226-230 in the resubmitted manuscript) has been significantly shortened:

"In the previous calculations, a charge ordered superlattice in the substrate is presumed, which exerts a classical superlattice Coulomb potential to graphene. However, this assumption should be re-examined. Moreover, besides the effects from the substrate to graphene, the feedback effects from graphene to the substrate should be discussed as well. Therefore, in this section, we study the coupled bilayer system as a whole, and treat the electrons in graphene layer and the substrate layer on equal footing."

4. S2: "We also provide separately six videos in other Supplementary Information" which cannot be found.

Reply: we sorry for failing to upload the videos to the submission system. Now we hope that the videos have been successfully uploaded this time together with the resubmitted manuscript.

5. Above (S65) 'less harsher'

Reply: we thank the reviewer for the careful reading of Supplementary Material. We have fixed this typo. Thanks a lot.

6. Below Fig. S11 'given by sing the charge'

Reply: again, we thank the reviewer for the careful reading of our manuscript. This typo has been fixed.

7. A few other obvious typos in the supplement such as 'from' rather than 'form'.

Reply: we have gone through the Supplementary Material for multiple times, and have fixed all the typos.

List of Significant Changes:

1. Fig. 4 in the main text is replaced by a new one, in which the condensation energy for electrons in the decoupled substrate is calculated using a fitting model determined by quantum Monte Carlo calculations, while the interlayer Coulomb energy is obtained from unrestricted Hartree-Fock calculations.
2. In Methods section, we add a few paragraphs (line 380 to line 390) and Eqs. (11)-(13) to describe the model of condensation energy for 2D electron gas, which is determined by fitting to quantum Monte Carlo data.
3. We have added a few sentences starting from line 276 of the main text:
"The energy difference between the EC state and Fermi-liquid (FL) state (condensation energy) as a function of the carrier density n is given by quantum Monte Carlo calculations in [55,56], as shown by the green line in Fig. 4(c). The condensation energy reaches zero when $n \approx 4.5 \times 10^{12} \text{ cm}^{-2}$ suggesting the transition from the EC to the FL state."
4. We have added the following sentences starting from line 286 of the main text:
"Specifically, with the separable wavefunction ansatz (Eq. (6)), the ground-state charge densities for the graphene layer and the EC layer are obtained from Hartree-Fock calculations, which are further used to estimate the interlayer Coulomb energy."
5. We have changed the sentences in line 117 to the following ones: *"...leading to two nearly degenerate insulating states, one is σ_z -sublattice polarized and the other is characterized by the order parameter $\tau_z \sigma_z$, where τ_z and σ_z denote the third Pauli matrix in the valley and sublattice space, respectively."*
6. We have added the following sentences starting from line 131: *"...This is because in Eq. (2), the interlayer Coulomb potential only applies to the situation of a single valley to accommodate the charge carriers in the substrate. In reality, there may be additional valley degeneracy in the substrate, which is crucial for the evolution of gap for $n_{\text{tot}} \rightarrow 0$. Although the valley degeneracy of the substrate does not change our results qualitatively, the theoretically calculated gap vs. n_{tot} fits to the experimental data of CrOCl-graphene heterostructure more precisely at low density once including the two-fold valley degeneracy of CrOCl (see Table I). The details are given in Supplementary Material (Fig. S11) [22]."*
7. We have added a new paragraph starting from line 156 of the main text in the resubmitted manuscript:
"Although it has been theoretically proposed that the magnetic proximity effect together with spin-orbit coupling could in principle give rise to topologically nontrivial states in graphene [46], it seems to be irrelevant to the graphene-insulator heterostructures considered in the present study. For example, in CrOCl-graphene device, no magnetic

hysteresis has been observed in graphene, and the measured Landau level degeneracy is still compatible with that of spin-valley degenerate Dirac cones [19]. Most saliently, the gap opening and the robust quantum Hall effect persist up to temperatures far above the Neel temperature of CrOCl (~14 K) [19]. Similarly, the magnetic proximity coupling was also reported to be negligible for CrI₃-graphene heterostructure [21]. Therefore, compared to the power-law decaying interlayer Coulomb coupling, the exponentially decaying magnetic proximity coupling may not play an important role in such charge-transfer graphene-insulator heterostructures."

8. We have added a new paragraph in main text starting from line 306 of the resubmitted manuscript:

"We note that the stabilizing effect of EC is not unique to band-aligned graphene-insulator heterostructures considered in this work. In general, it only requires the presence of another (tunable) gapped state exhibiting non-uniform charge distribution atop of the EC. For example, remarkably robust EC state has been observed in a bilayer system consisting of two monolayer MoSe₂ separated by hexagonal boron nitride [57], which was also argued to be stabilized by the interlocking of the EC states in the two layers."

9. We have added a new equation (Eq. (3)) and a new paragraph discussing the universality of our results starting from line 89 in the resubmitted manuscript.

10. We have added the following sentence starting from line 73 of the resubmitted manuscript: *"Such a continuum-model description is adopted throughout the paper given that $L_s \gg a$ is always fulfilled for low carrier density $< \sim 10^{13} \text{ cm}^{-2}$, with $L_s \sim 1/\sqrt{n}$ for the EC state. "*

11. We have added the zero-energy convention in the caption of Fig. 2: *"Zero energies in (b) and (c) are defined as the chemical potentials for $\nu = 0$ and $\nu = -0.003$, respectively. "*

12. We have added the following sentences starting from line 253 in the main text:

"This is further confirmed by directly calculating the orbital projected band structures of a commensurate supercell of CrOCl-graphene heterostructure based on density functional theory. It turns out that the Dirac cone in such heterostructure supercell stems almost 100% from carbon p_z orbitals of graphene (see Sec. S7 of Supplementary Material), which clearly indicates the absence of interlayer hybridization (hopping)."

We have also added a new subsection Sec. S7 "Band structure of graphene-CrOCl heterostructure" in Supplementary Material, showing the orbital projected band structure of one particular commensurate supercell of graphene-CrOCl heterostructure (Fig. S15)

13. We have also included the following sentences starting from line 272 of the main text, which briefly discuss the possibility of interlayer excitonic insulator in valley-matched

graphene-insulator heterostructures:

"Nevertheless, the interlayer excitonic insulator state consisted of Dirac electrons (holes) and quadratically dispersive holes (electrons) is possible in valley-matched graphene-insulator heterostructures, such as those consisted of graphene and transition metal dichalcogenides with the band extrema at K points. We leave this for future study."

14. We have added the following sentences starting from line 194: *"On the one hand, the band crossing moving away from Γ_s is of accidental nature, which is generally avoided unless the lattice parameters are at some fine-tuned values. On the other hand, the Dirac point at Γ_s remains stable as protected by $C_{2z}T$ symmetry."*

15. We provide six videos as Supplementary Material illustrating the evolution of the band structures and Berry curvatures as a function of the lattice anisotropy parameter.

16. We have added optical images of the CrOCl-graphene heterostructure device, as well as the detailed explanation of the measurement setup in Sec. S8. We have added the following sentence starting from line 454 in Methods section: *"More details about the device configuration, measurement set up, and sample quality can be found in Sec. S8 of Supplementary Material."*

17. We have added five more references:

Ref. 33, J. E. Drut and T. A. Lahde, Phys. Rev. Lett. 102, 026802 (2009)

Ref. 46, Z. Qiao, W. Ren, H. Chen, L. Bellaiche, Z. Zhang, A. H. MacDonald, and Q. Niu, Phys. Rev. Lett. 112, 116404 (2014)

Ref. 56, F. Rapisarda and G. Senatore, Australian journal of physics 49, 161 (1996)

Ref. 57, Y. Zhou, J. Sung, E. Brutschea, I. Esterlis, Y. Wang, G. Scuri, R. J. Gelly, H. Heo, T. Taniguchi, K. Watanabe, et al., Nature 595, 48 (2021)

Ref. 63 J. R. Trail, M. D. Towler, and R. J. Needs, Phys. Rev. B 68, 045107 (2003)

18. Since two students, Yaning Wang and Zhongqing Guo, have made important contributions to this work during the past three months, we have added these two people as our coauthors.

All of these changes have been marked in red in the resubmitted manuscript.

REVIEWERS' COMMENTS

Reviewer #1 (Remarks to the Author):

The authors have sufficiently addressed my concerns in the revised manuscript, and I now recommend its publication in Nature Communications.

Reviewer #2 (Remarks to the Author):

The authors of "Synergistic Correlated States and Nontrivial Topology in Coupled Graphene-Insulator Heterostructures" have comprehensively addressed all of my previously stated concerns and have provided satisfactory responses, complete with additional calculations and relevant references, to inquiries made by other referees. Therefore, I recommend it for publication in Nature Communications.